# EMBO
*reports*

# Cytoplasmic control of Rab family small GTPases through BAG6

Toshiki Takahashi[1], Setsuya Minami[1], Yugo Tsuchiya[1], Kazu Tajima[1], Natsumi Sakai[1], Kei Suga[2,3], Shin-ichi Hisanaga[4], Norihiko Ohbayashi[5], Mitsunori Fukuda[6] & Hiroyuki Kawahara[1,*]

## Abstract

Rab family small GTPases are master regulators of distinct steps of intracellular vesicle trafficking in eukaryotic cells. GDP-bound cytoplasmic forms of Rab proteins are prone to aggregation due to the exposure of hydrophobic groups but the machinery that determines the fate of Rab species in the cytosol has not been elucidated in detail. In this study, we find that BAG6 (BAT3/Scythe) predominantly recognizes a cryptic portion of GDP-associated Rab8a, while its major GTP-bound active form is not recognized. The hydrophobic residues of the Switch I region of Rab8a are essential for its interaction with BAG6 and the degradation of GDP-Rab8a via the ubiquitin-proteasome system. BAG6 prevents the excess accumulation of inactive Rab8a, whose accumulation impairs intracellular membrane trafficking. BAG6 binds not only Rab8a but also a functionally distinct set of Rab family proteins, and is also required for the correct distribution of Golgi and endosomal markers. From these observations, we suggest that Rab proteins represent a novel set of substrates for BAG6, and the BAG6-mediated pathway is associated with the regulation of membrane vesicle trafficking events in mammalian cells.

**Keywords** BAG6; membrane trafficking; Rab GTP-binding proteins; Rab8a; ubiquitin-proteasome system

**Subject Categories** Membrane & Intracellular Transport; Post-translational Modifications, Proteolysis & Proteomics

## Introduction

Rab family small GTPases are critical for maintaining the function and architecture of cytoplasmic organelles by controlling membrane trafficking between them [1–5]. The human genome encodes more than 60 Rab protein genes [1,5], and each Rab protein possesses its own function corresponding to the pathway that it regulates [5–7]. Indeed, together with SNARE proteins, which mediate vesicle docking and fusion [8,9], distinct Rab GTPases localize to different membrane compartments in order to control the specificity and directionality of membrane trafficking pathways [5]. Rab8, for example, plays an important role in the tubulovesicular trafficking of cargo proteins destined for the plasma membrane through recycling endosomes (RE) en route from the *trans*-Golgi network (TGN) [10–15].

Rab proteins control membrane trafficking by cycling between active GTP-bound and inactive GDP-bound forms that differ primarily by the conformation of two nucleotide-surrounding loops known as "Switch" regions [16]. These conformational changes are critical, since Rab proteins in the GTP-bound form at the membrane become competent to bud and recruit vesicles to target destinations with specific effector interactions [17–19]. On the other hand, the GDP-bound conformation stimulates binding to distinct factors that facilitate dissociation from the target membrane, and retroverting Rabs to departure organelles [20,21]. Thus, Rab GTPases shuttle between the cytosol and membranes. GDP-bound forms of Rab proteins in the cytosol are prone to aggregation due to the exposure of hydrophobicity in their C-terminal membrane anchor motif with lipophilic geranylgeranyl groups (consisting of 20-carbon isoprenoid groups) [22], as well as in the unfolded Switch I region [16,18,23,24]. Although several chaperone-like factors, including GDP-dissociation inhibitors (GDIs), Rab escort protein (REP) and RABIF/MSS4 (mammalian suppressor of yeast Sec4), have been reported to help prevent the aggregation of Rab proteins [22,23,25–27], the cytoplasmic machinery that determines the fate of the GDP-bound form of specific Rab species remains largely unknown.

BAG6 (also called BAT3 or Scythe) is a chaperone/holdase that contains a ubiquitin-like domain and interacts with aggregation-prone hydrophobic polypeptides and escorts them to the degradation machinery [28–35]. BAG6 possesses an intrinsic affinity for the hydrophobic residues of client proteins [28,30–32,36,37] and captures newly synthesized polypeptides by means of their exposed hydrophobic patches concomitant with or after their release from

1  Laboratory of Cell Biology and Biochemistry, Department of Biological Sciences, Tokyo Metropolitan University, Tokyo, Japan
2  Department of Cell Physiology, Kyorin University School of Medicine, Mitaka, Japan
3  Department of Chemistry, Kyorin University School of Medicine, Mitaka, Japan
4  Laboratory of Molecular Neuroscience, Department of Biological Sciences, Tokyo Metropolitan University, Tokyo, Japan
5  Department of Physiological Chemistry, Faculty of Medicine, University of Tsukuba, Tsukuba, Japan
6  Department of Integrative Life Sciences, Graduate School of Life Sciences, Tohoku University, Sendai, Japan
   *Corresponding author. Tel: +81-42-677-1111; E-mail: hkawa@tmu.ac.jp

the ribosome [30,31,38], which improves their solubilization, assembly, and/or degradation efficiency [31,32,35,36,39]. BAG6 also captures and shields the exposed transmembrane domain (TMD) of newly synthesized TMD proteins in the cytosol for subsequent degradation if their proper biogenesis has failed [31,35,39–43]. Thus, a series of studies have shown crucial roles for BAG6 in the quality control of newly synthesized TMD proteins [31–33,35,40,41,43,44]. However, it is unknown how BAG6 contributes to the physiological aspects of other sets of membrane proteins, such as phospholipid-anchored small GTPases.

In this study, we found that BAG6 knockdown induced the abnormal localization of several Golgi/endosomal marker proteins, as well as defects in membrane protein sorting to the plasma membrane in human and hamster cells. Subsequent analysis revealed the physical association of BAG6 with Rab8a. BAG6 showed a clear preference for the GDP-bound form of Rab8a, while its active form was scarcely recognized. Although the major population of Rab8a, that is, the GTP-bound form, was highly stable, GDP-bound Rab8a was degraded rapidly by the BAG6- and proteasome-mediated pathway, and the accumulation of the GDP-bound form caused defects in intracellular membrane trafficking. We show evidence that the hydrophobic residues of the Switch I region, which have been reported to be exposed in its GDP-bound form, are essential for Rab8a binding with BAG6, and determined the instability of cytoplasmic Rab8a. Finally, BAG6 bound not only with Rab8a but also with a functionally distinct set of Rab family proteins. From these observations, we suggest that BAG6 possesses an unexpected but critical function in maintaining the integrity of membrane trafficking events by limiting the excess accumulation of GDP-bound forms of Rab small GTPases.

## Results

### BAG6 deficiency induces defects in the distribution of endosomal proteins

In BAG6-suppressed cells, we noticed defects in the intracellular localization of various TMD proteins. For example, the transferrin receptor (TfnR), an endosomal protein that is delivered from the TGN to tubular RE [14,45,46], was found scattered around a peripheral region in the cytoplasm of BAG6-suppressed HeLa cells (Fig 1A, right panel). In contrast, the majority of TfnR in control knockdown (or non-treated) cells was distributed in cytoplasmic tubular structures, a characteristic of RE proteins reported previously (Fig 1A, left panel) [12,47]. The distribution of TfnR in BAG6-suppressed cells appeared similar, if not identical, to the case reported for Rab8a knockdown (Fig 1A, center panel), a key protein in the endosomal localization/trafficking of TfnR [48].

In the case of the 12-pass TMD protein Patched 1 (Ptc1), it was reported that the majority of this endosomal protein can be detected in cytoplasmic vesicular structures [49], and our immunostaining in control cells showed a similar pattern of typical Ptc1 signals (Fig 1B, left panel). On the contrary, the distribution of Ptc1 was altered greatly to biased perinuclear compartments and reduced cytoplasmic vesicular dots in BAG6-defective cells (Fig 1B, right panel; see also Fig EV1A). The defective distribution of Ptc1 was also observed following Rab8a depletion (Fig EV1B), which was a similar

phenotype to that observed in BAG6-depleted cells (Figs 1B and EV1A). The perinuclear accumulation of the Ptc1 signal in *BAG6*-suppressed cells was not derived from protein aggregates since the immunosignal was negative for polyubiquitin staining (Fig EV1A, indicated by arrowheads), a marker for cytoplasmic protein aggregates (Appendix Fig S1A, indicated by an arrow) [30,50]. Ptc1 was reported to be down-regulated by the lysosomal pathway with a relatively short half-life [51,52]. In accordance with these reports, the degradation of Ptc1 protein was inhibited by leupeptin, a lysosomal protease inhibitor, while the proteasome inhibitor epoxomicin had little effect (Appendix Fig S1B). While BAG6 was reported to be linked with proteasome-mediated degradation [31,42], BAG6 knockdown clearly stabilized Ptc1 protein and stimulated its accumulation (Fig 1C and D). In addition, three independent small interfering RNAs (siRNAs) targeting Rab8a also resulted in the accumulation of Ptc1 protein (Fig 1C), suggesting that Rab8a-mediated vesicular trafficking controls Ptc1 stability. These results imply that endogenous BAG6 protein may have a novel role in endosomal/lysosomal protein sorting.

As both TfnR and Ptc1 are typical endosomal proteins, we examined the distribution of Rab7 as a marker for late endosomes [53]. We found that Rab7 localization was altered in BAG6 knockdown cells (Appendix Fig S1C). In contrast, the staining of the ER marker calnexin in BAG6 knockdown cells did not show any obvious differences compared with control cells (Fig EV1C and D). Therefore, BAG6 may have a role in controlling the membrane trafficking of endosomes, either directly or indirectly.

### BAG6 physically interacts with Rab8a

To gain insight into the mechanism by which BAG6 regulates the localization of endosomal proteins, we analyzed mass spectrometry-based comprehensive human protein–protein interaction network databases [54,55]. This analysis suggested potential interactions between Rab8a and BAG6, as well as with other BAG6-accessory proteins such as Ubl4a [38,56–58] (Fig 2A). Rab8a was originally described to regulate the membrane trafficking of newly synthesized endosomal proteins to the basolateral and apical surfaces [10,15,59], and Rab family GTPases are master regulators of distinct steps of intracellular vesicle trafficking in eukaryotic cells [12–14,47,60]. Therefore, we examined whether BAG6 was physically associated with Rab family proteins. As shown in Fig 2B, Rab8a was co-precipitated by BAG6 pull-down from HeLa cell extracts, as efficiently as in the case of the model BAG6 client protein luciferase (Lc)-CL1 [37]. In contrast, related Rab family small GTPases, such as Rab7 and Rab11a, did not co-precipitate with BAG6 under identical conditions (Fig 2B), suggesting strict binding specificity. We also confirmed that endogenous Rab8a protein was associated with BAG6 (Fig EV2A). The binding of BAG6 with Rab8a was also confirmed by Flag-Rab8a immunoprecipitation experiments (Fig EV2B). Collectively, these results suggest that the BAG6 complex associates intimately with specific Rab family proteins.

### BAG6 recognizes Rab8a in a nucleotide-specific manner

The Rab8a (Q67L) mutant lacks intrinsic GTPase activity and is thus constitutively active as the GTP-bound form, whereas the Rab8a (T22N) mutant is an inactive species that dominantly binds with

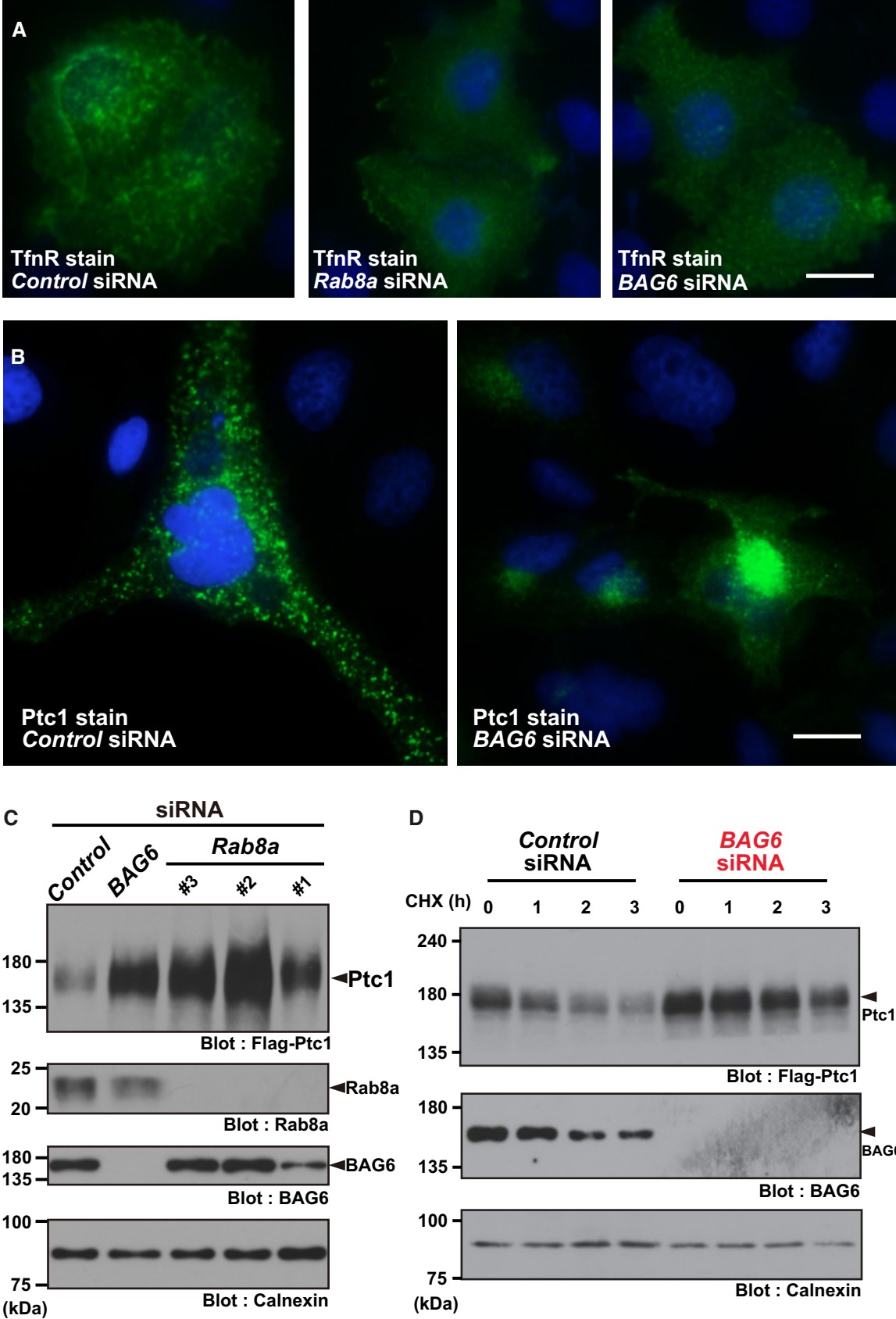

**Figure 1.**

Figure 1.  Defective distribution of endosomal proteins in BAG6-suppressed cells.

A    At 72 h after transfection with siRNA duplexes for *BAG6* or control siRNA (10 nM each), the intracellular localization of TfnR in HeLa cells was examined (shown as green). Nuclear DNA was stained with Hoechst 33342 (shown as blue). Control knockdown (left panel), Rab8a knockdown (center panel), and BAG6 knockdown (right panel). Efficacy of endogenous BAG6 knockdown in HeLa cells was verified by Western blot experiments (see Fig EV1D). Scale bar: 10 μm.

B    Intracellular localization of Ptc1 (green) in HeLa cells. Nuclei were stained with Hoechst 33342 (shown as blue). See also Fig EV1A and B. Scale bar: 10 μm.

C, D    Knockdown of Rab8a (with *Rab8a* siRNA#1, #2, and #3) or BAG6 (with *BAG6* siRNA#1) stimulated the accumulation and stabilization of Ptc1 protein in HEK293 cells. See also Appendix Fig S1B.

Source data are available online for this figure.

GDP [11]. We found that Rab8a (Q67L) showed reduced interactions with BAG6 compared with wild-type (WT) Rab8a (Fig 2C–E). In contrast, greatly enhanced binding with BAG6 was observed in the case of the GDP-bound inactive mutant Rab8a (T22N; Fig 2C–E). Binding of BAG6 with WT Rab8a or the T22N mutant was enhanced by the addition of protease inhibitors such as MG-132 (Figs 2C and EV2B).

Rab GTPases must be converted from their GDP-bound inactive state to their GTP-bound active conformation, a process stimulated by a guanine nucleotide exchange factor (GEF) [61–64]. To examine whether the nucleotide-binding state of Rab8a is a critical factor for BAG6 recognition, we suppressed the endogenous GEF for Rab8a, namely, Rabin8/Rab3IP [19,48,65–67]. The results showed the greatly enhanced association of BAG6 with WT Rab8a under Rabin8 knockdown (Fig 2F), further supporting the idea that BAG6 predominantly recognizes the GDP-bound inactive form of Rab8a protein as its client protein.

BAG6 has been reported to be a cytoplasmic triage factor [68] and is critical for both tail-anchored (TA) protein insertion [38,56] and the degradation of failed insertion products [31]. BAG6 possess a strong preference for binding to the exposed hydrophobic residues of membrane proteins in the cytosol [31,35,42,43]. Since Rab8a protein is post-translationally modified by highly lipophilic geranylgeranylation at the C-terminal Cys[204] residue, which is essential for its membrane anchoring (Fig 3A), and since Rab GTPases exist in both cytoplasmic (GDP-bound form) and membrane-bound (GTP-bound form) pools [3,16], we suspected that BAG6 might recognize such exposed hydrophobic residues of the un-embedded lipophilic group when it dissociates from membranes as the GDP-bound form. To examine whether C-terminal geranylgeranylation is essential for the association of Rab8a (T22N) with BAG6, we mutated Cys[204] to a serine residue (designated C204S mutant Rab8a, Fig 3A). As shown in Fig EV2C, C204S mutant Rab8a protein localized exclusively in the soluble cytoplasmic fraction, in contrast to its WT counterpart that was located in the insoluble membrane fraction. Unexpectedly, the C204S mutation did not affect BAG6 binding to Rab8a (Fig 3B). In accordance with this notion, the 100 C-terminal residues of Rab8a were dispensable for its association with BAG6 (Fig 3C, N100). In contrast, deleting 100 N-terminal residues from Rab8a (Rab8a ΔN100) abolished its interaction with BAG6, even in the presence of MG-132 (Fig 3C), suggesting that the N-terminal GTPase domain is critical for BAG6 recognition.

## GDP-bound form of Rab8a is unstable and is targeted for ubiquitin-mediated degradation

Although cycloheximide (CHX) chase experiments confirmed that WT Rab8a was highly stable in HeLa cells (Fig EV2D), the association between BAG6 and Rab8a was greatly enhanced in the presence of a protease inhibitor (Figs 3B, and EV2A, B and D), suggesting a possible link between their interactions and protein degradation events. Since the BAG6-Rab8a interaction was mediated by the N-terminal 465 residues of BAG6 (Fig 3D), which is an essential region for recognizing a variety of hydrophobicity-exposed substrates for ubiquitin-mediated degradation [30–32,36,37,69], we suspected that the GDP-bound form of Rab8a would be unstable. Strikingly, Rab8a (T22N) was unstable with a half-life of < 3 h, while the GTP-bound mutant Q67L was highly stable (Figs 3E and EV2F). The instability of the T22N mutant protein was not perturbed by the C204S mutation (Fig 3F), suggesting that the elimination of the Rab8a GDP-bound form can be mediated by the cytoplasmic degradation machinery. In accordance with this idea, the 100 residue N-terminal fragment of the Rab8a GTPase domain was highly sensitive to MG-132 (Fig 3G) with high affinity for BAG6 (Fig 3C). In contrast, the ΔN100 truncated fragment was insensitive to MG-132 and had decreased interactions with BAG6 (Fig 3C and G).

The degradation of full-length Rab8a (T22N) was blocked by MG-132 (Z-Leu-Leu-Leu-CHO), but not by the structurally related lysosome inhibitor leupeptin (Ac-Leu-Leu-Arg-CHO; Fig EV2E), suggesting that the proteasome is responsible for its degradation [70,71]. In accordance with this notion, a large amount of polyubiquitin chains co-precipitated with Rab8a (T22N) in the presence of proteasome inhibitors (Fig 3H and Appendix Fig S2A). Highly stable Rab8a (Q67L) and WT Rab8a (a mixture of predominant GTP-bound and minor GDP-bound forms) were subjected to low-level polyubiquitination (Fig 3H and Appendix Fig S2B) in contrast to the case for Rab8a (T22N). In addition, polyubiquitination was not observed for N-terminally deleted Rab8a (ΔN100, Appendix Fig S2C), which did not show BAG6 binding (Fig 3C). Finally, to examine whether Rab8a (T22N) protein itself is ubiquitinated directly, the cells were hot lysed by treatment with 1% sodium dodecyl sulfate (SDS) at 90°C, and Rab8a was immunoprecipitated. Such denaturation of Rab8a protein did not affect its co-precipitation of polyubiquitin chains (Appendix Fig S2D), providing evidence that Rab8a (T22N) is covalently modified with ubiquitin chains.

## Exposed hydrophobicity of the Switch I region in Rab8a is critical for BAG6 recognition and instability

How BAG6 recognizes Rab8a is an important issue to understand the control mechanism of the Rab8a cycle. Since the GDP-bound form of Rab8a specifically associated with BAG6 (Fig 2C), BAG6 binding with Rab8a could be influenced mostly through structural features in the GTPase domain induced by the nucleotide-binding cycle. It has been reported that guanine nucleotide-binding induces

structural changes of the GTPase domain of Rab family proteins to interact with specific effectors [16]. Two Switch regions, namely, Switch I and II, are the only regions within the GTPase domain that change conformation between the GTP- and GDP-bound forms (crystal structures of the active and inactive forms of Rab8, 4LHW, and 4LHV, respectively, Appendix Fig S3A and B) [16,66]. Thus, we paid particular attention to the hydrophobic residues within the Switch regions of Rab8a, since the N-terminal region of BAG6 has been shown to bind preferentially with the hydrophobic stretches of its client proteins [37,42]. Previous structural studies revealed that

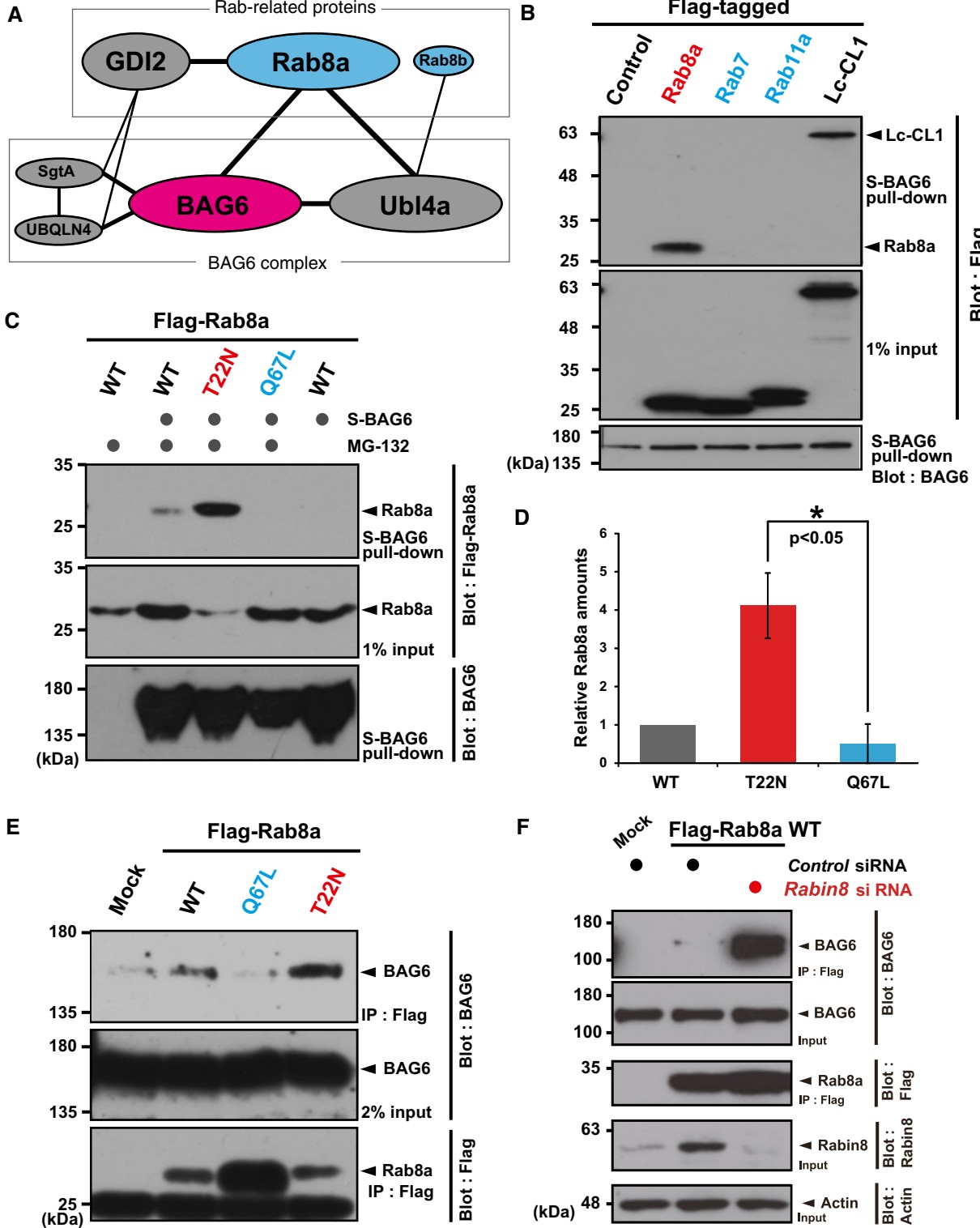

**Figure 2.**

**Figure 2.   BAG6 physically interacts with Rab8a in a nucleotide-binding specific manner.**

A    Protein interaction network suggested by public databases.

B    BAG6 protein co-precipitated Rab8a, while neither Rab7 nor Rab11a was co-precipitated with BAG6. Flag-tagged Rab8a, Rab7, Rab11a, and luciferase-CL1 (Lc-CL1; a positive control) were expressed in HeLa cells and the cells were treated with 10 μM MG-132 for 4 h. Flag-immunoprecipitates were blotted with anti-BAG6 and anti-Flag antibodies, respectively. Note that all cells used were treated with 10 μM MG-132 for 4 h.

C    S-tagged BAG6 pull-down efficiently co-precipitated Rab8a (T22N), a GDP-bound mutant, while BAG6 scarcely co-precipitated Rab8a (Q67L), a constitutively active mutant. Co-precipitation of Rab8a WT with BAG6 was used as a standard. MG-132 (10 μM) was included in the cell culture for 4 h, as indicated. Note that the T22N mutant protein was expressed at lower levels than either WT or Q67L, and that this was partly due to increased degradation, as will be shown later. S-BAG6 stands for N-terminally S-tagged BAG6 protein (see the Materials and Methods).

D    Anti-Flag signals in (C) were quantified, and relative signal intensities are presented. The value of the WT Rab8a signal with MG-132 was defined as 1.0. Note that all signal intensities of Flag-tag were normalized by that of the Rab8a input signals. The graph represents the mean ± standard deviation (SD) calculated from three independent biological replicates. An asterisk indicates $P < 0.05$ (Student's *t*-test).

E    A series of Flag-Rab8a mutants were immunoprecipitated and quantified the amount of endogenous BAG6 that were co-precipitated with Flag-Rab8a. Note that all cells used were treated with 10 μM MG-132 for 4 h.

F    Deficiency of Rabin8, a GEF for Rab8a, enhanced the physical interaction between BAG6 and WT Rab8a proteins. Flag-tagged WT Rab8a was expressed in *Rabin8* siRNA-treated cells, and Flag-immunoprecipitates were probed with an anti-BAG6 antibody. Note that all cells used were treated with 10 μM MG-132 for 4 h.

Source data are available online for this figure.

several conserved hydrophobic residues in the Rab8a Switch I region, namely, Ile[38], Ile[41], and Ile[43], are exposed to the cytoplasmic surface when it is bound to GDP, while these residues are hidden inside the molecule in the GTP-bound form (Fig 4A and B and Appendix Fig S3A and B) [16,66]. To examine whether these hydrophobic residues in Rab8a are critical for BAG6 binding, we substituted them with hydrophilic serine residues and designated this mutant as Rab8a (T22N-3IS) (Fig 4A and B). When we examined its interaction with BAG6, the Rab8a (T22N-3IS) mutant failed to co-immunoprecipitate with BAG6 (Fig 4C), suggesting that the exposed hydrophobic residues of the Switch I region in its GDP-bound form are essential for Rab8a binding to BAG6. In addition, the "8a-7a chimera" T22N protein with Rab7-type Switch I sequence showed reduced affinity with BAG6 compared to the case for Rab8a (T22N) (Fig EV3A and B).

Since the Rab8a (T22N-3IS) mutant was shown to lose its interaction with BAG6 (Fig 4C), we examined whether this Switch I mutant was stabilized in cells. As shown in Fig 4D, the 3IS mutation greatly increased the stability of this protein compared to Rab8a (T22N) (Figs 4D and EV2G). Furthermore, the Rab8a (T22N-3IS) mutant protein was not polyubiquitinated, in contrast to the case for Rab8a (T22N) (Fig 4E). These observations suggest that the hydrophobic residues in the Switch I region are responsible for the instability of cytoplasmic Rab8a, and BAG6 might target such a GDP-bound form of Rab8a through these exposed hydrophobic residues for ubiquitin-mediated degradation.

To estimate directly the impact of BAG6 on the instability of the GDP-bound form of Rab8a, we compared its stability in the presence or absence of *BAG6* siRNA. We found that *BAG6* knockdown stimulated Rab8a (T22N) accumulation and increased its stability (Fig 5A and B). Furthermore, polyubiquitination of Rab8a (T22N) was decreased in *BAG6*-depleted cells (Fig 5C). Collectively, these results indicate that endogenous BAG6 is necessary for the ubiquitin-mediated elimination of cytosolic (GDP-bound form) Rab8a. Since *BAG6* knockdown did not show complete stabilization of Rab8a (T22N), as observed for the Rab8a (T22N-3IS) mutant protein (Figs 4D and 5A), a partly redundant degradation pathway may exist, which remains to be determined.

It has been suggested that the forced expression of Rab8a (T22N) inhibits the transformation of macropinosomes into tubules [15]. In accordance with this, the excess accumulation of the inactive form of Rab family GTPase proteins tends to cause defects in intracellular membrane trafficking events [15,25,72–78]. Accordingly, we tested whether the accumulation of the GDP-bound form of Rab8a might affect the distribution of endosomal proteins. As shown in Fig 5D, the forced expression of Rab8a (T22N) protein in HeLa cells abolished the typical tubular structures of TfnR distribution, similar to the case of BAG6 depletion (Fig 1A). Therefore, the insufficient elimination of Rab8a in its GDP-bound form can induce the defective distribution of recycling endosomes.

## BAG6 possesses a distinct preference for multiple Rab species

The human genome encodes more than 60 Rab family proteins, whose functions are specifically controlled at distinct steps of membrane vesicle trafficking [1–5]. Although both Rab7 and Rab11a co-immunoprecipitated scarcely with BAG6 (Fig 2B), we were interested in examining whether another set of Rab family proteins may possess an affinity for BAG6 as in the case of Rab8a, since the primary sequence of the Rab8a GTPase domain recognized by BAG6 is fairly well conserved in other Rab family proteins (Fig 4A and Appendix Fig S4). To address this possibility, we examined the interactions of BAG6 with a series of Rab family proteins. Comprehensive immunoprecipitation analyses revealed that BAG6 showed a distinct preference to WT Rab family proteins (Fig 6A and B). Rab proteins, including Rab9b, Rab1b, Rab13, Rab2b, Rab6b, Rab3a, Rab10, Rab15, Rab1a, and Rab8a, were efficiently co-precipitated with BAG6 (Fig 6B). These observations implied that BAG6 binds not only to Rab8a but also to a functionally distinct set of Rab family GTPases (here designated as the "BAG6-phile" Rab family). In contrast to the case of these BAG6-phile Rab family proteins, only a small amount of BAG6 association was observed with Rab14, Rab6a, Rab4, Rab5a, and Rab5b (Fig 6B). Some of the major Rab family proteins, such as Rab7a, Rab11a, Rab11b, and Rab12, did not bind with BAG6 (Fig 6B; see also Fig 2B). Amino acid sequence alignments of the Switch I regions (Appendix Fig S4) and Kyte-Doolittle hydrophobicity plots (Appendix Fig S5) suggest that BAG6-phile Rab species tended to possess a higher hydrophobicity peak in the Switch I region

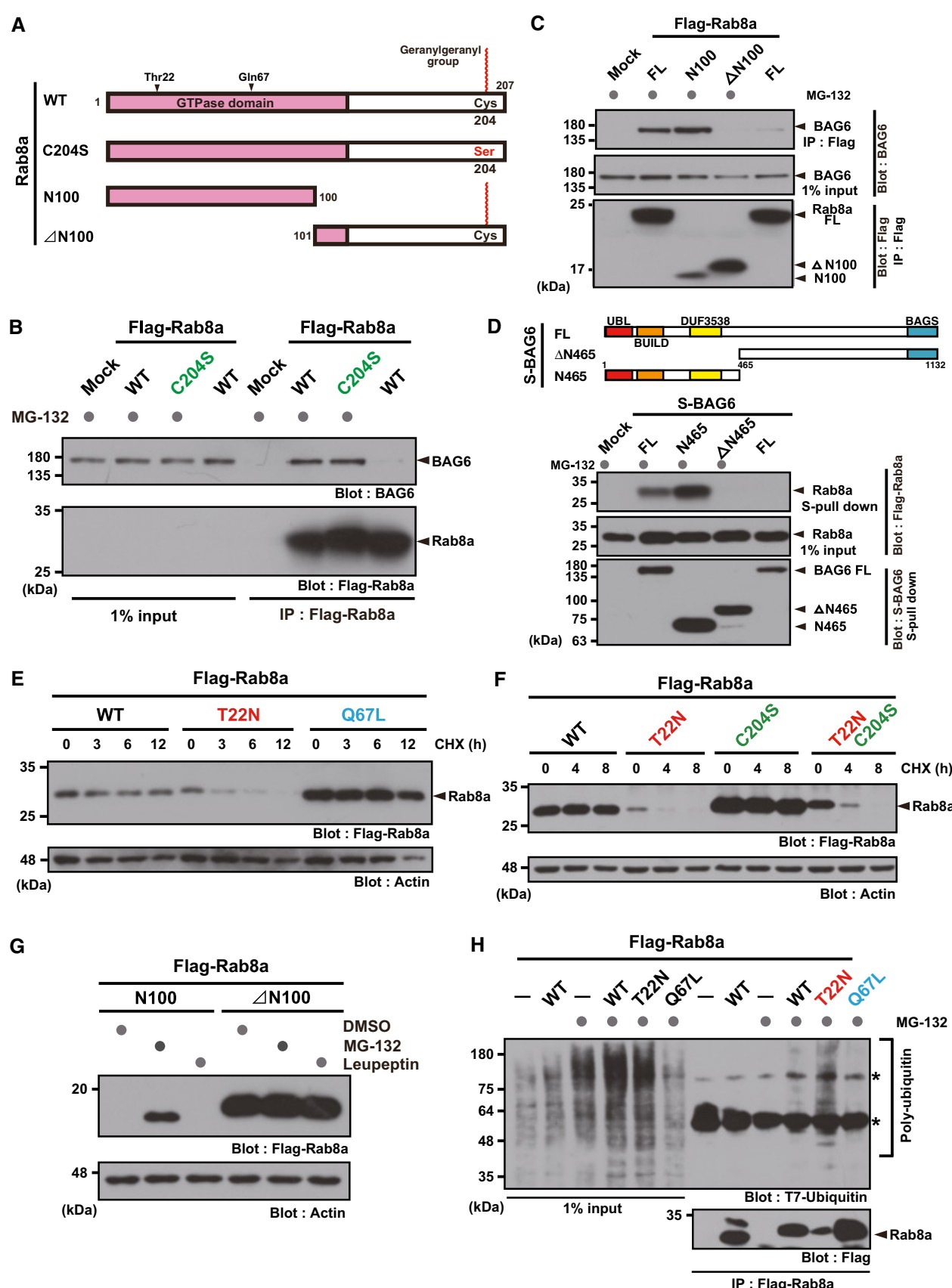

**Figure 3.**

**Figure 3.  GDP-bound form of the Rab8a GTPase domain is critical for its ubiquitin-mediated degradation.**

A    Schematic representation of Rab8a protein. To prevent C-terminal geranylgeranylation, the Cys²⁰⁴ residue was substituted with a serine residue. Alternatively, two truncated proteins (N100 and ΔN100 fragments of Rab8a) were prepared for this experiment.

B    C-terminal geranylgeranylation of Rab8a was dispensable for BAG6 recognition.

C    The N-terminal GTPase domain of Rab8a was essential for BAG6 recognition. A series of Flag-tagged truncated fragments of WT Rab8a were expressed in HeLa cells with S-tagged BAG6 and treated with (+) or without (−) protease inhibitors for 4 h. Flag-Rab8a substrates were immunoprecipitated and probed with an anti-BAG6 antibody.

D    Hydrophobicity recognition domain of BAG6 (N465) was critical for Rab8a recognition. A schematic representation of the BAG6-truncated proteins used in this experiment is shown in the upper panel. Numbers denote the corresponding amino acids of mammalian BAG6. Positions of UBL, BUILD, and DUF3538 domains, which are all linked to hydrophobicity recognition by BAG6 [37], are indicated. WT, wild-type; ΔN465, N-terminal 465 residues-deleted mutant; and N465, C-terminal 689 residues-deleted mutant.

E, F    CHX chase experiments show that GDP-bound Rab8a (T22N) was a highly labile protein, while the GTP-bound active mutant (Q67L) was a stable protein (E). Instability of the T22N mutant was not perturbed by the C204S mutation (F). Actin was used as a loading control.

G    N-terminal 100 residue fragment of the Rab8a GTPase domain was sensitive to the proteasome inhibitor, while the ΔN100 fragment was not.

H    Rab8a (T22N) was polyubiquitinated in the presence of the proteasome inhibitor. Asterisks indicate non-specific bands.

Source data are available online for this figure.

compared with Rab proteins with weak BAG6 interactions. We did not observe any correlation between the retrograde/anterograde trafficking function of Rab proteins and the order of BAG6 affinity. These observations support the idea that BAG6 associates not only with Rab8a but also with multiple, but specific, Rab family proteins, and BAG6 might have pleiotropic functions in membrane trafficking events.

## BAG6 is required for multiple aspects of membrane trafficking events

Since multiple Rab family proteins are potential targets of BAG6 (Fig 6), BAG6 might influence several aspects of membrane sorting events. We found that BAG6 knockdown induced the abnormal distribution of the *trans*-Golgi marker syntaxin 6 (Stx6) in Chinese hamster ovary (CHO) cells (Fig 7A, shown as green; see also Fig EV4A and B using different siRNA target sequences to Fig 7A). The abnormally dispersed distribution of Stx6 was also observed in Rab8a-depleted CHO cells, which was nearly indistinguishable to the case in BAG6-depleted cells (Fig EV4B). The localization of other Golgi markers, namely, *cis*-Golgi matrix protein GM130 and ER-Golgi SNARE protein GS28, was also affected by BAG6 depletion (Fig 7B, shown as green for GS28 and red for GM130; see also Fig EV4A and C using different siRNA target sequences to Fig 7B). In control knockdown cells, GS28 and GM130 co-localized in a specific perinuclear compartment with a biased distribution, a typical pattern of the Golgi apparatus (Fig 7B, left panel). In BAG6 knockdown cells, however, their signals were separated with less co-localization throughout the perinuclear region of the cytoplasm (Fig 7B, right panel; compare the red and green signals), a similar phenotype to that observed in Rab8a-depleted cells (Fig EV4C). These observations suggest that not only endosomes (Fig 1) but also the Golgi apparatus are affected severely by BAG6 depletion. In addition, these observations indicate that the effects of *BAG6* siRNAs on organelles were conserved in different species, namely, humans (Figs 1 and EV1A) and hamsters (Figs 7A and B, and EV4), with their respective unique double-stranded RNA sequences (see Materials and Methods).

Since the Golgi apparatus is an organelle where glycosyl modification of membrane proteins occurs, we examined whether the glycosylation of IL-2Rα [42], a single-pass TMD protein, is affected by BAG6 knockdown in HeLa cells. As shown in Fig 7C, no defects in the glycosylation status of IL-2Rα protein were observed. Thus, defects in Golgi distribution seem not to be accompanied by general defects in the glycosylation of Golgi cargo proteins.

The Golgi apparatus is also known to function as a critical hub for glycoprotein sorting from the ER to the plasma membrane. Therefore, we evaluated the effect of BAG6 knockdown on the amount of cell surface glycoproteins, which was assessed using fluorescence-labeled lectin as a probe (Fig 7D). As a control, knockdown of SRP54 protein, a central subunit of the signal recognition particle [42,79], resulted in the disruption of the supply of glycoproteins to the cell surface (Fig 7E and Appendix Fig S6), supporting the validity of this assay to quantify the cell surface presentation of newly synthesized membrane proteins. With this system, we found that cell surface glycoproteins were decreased significantly in BAG6 knockdown cells (Fig 7E and F). These findings suggest a crucial role for endogenous BAG6 in glycoprotein transport to/from the plasma membrane via the intracellular vesicle trafficking pathway.

## Discussion

This study supports the idea that members of the small GTPase Rab family are one set of targets of BAG6 for controlling membrane trafficking events (Fig EV5). Since the BAG6 complex is also known as a key component of the TA protein insertion pathway, which mediates the biogenesis of SNARE proteins [38,56,68,80], this suggests that some of the observed trafficking defects might be due to SNAREs in addition to Rabs. Such a possibility obviously needs to be investigated in future studies.

It was curious for us that BAG6-Rab8a co-immunoprecipitation largely depended on the presence of proteasome inhibitors, in spite of the fact that both BAG6 and Rab8a are highly stable proteins. In-depth analysis revealed that BAG6 exclusively targeted a cryptic portion of the GDP-associated labile form of Rab8a protein (Figs 2C–F and EV5), while its major GTP-bound form (a very stable protein) was not recognized as a client. Importantly, the folding of Rab family GTPases is dependent on the bound

nucleotide [16,66], and we provided evidence that Rab8a (GDP-bound form) showed remarkable instability *in vivo* (Fig 3E and F). We also found that ubiquitination of Rab8a (GDP-bound form)

was governed by BAG6 (Figs 5C and EV5). Indeed, the exposed hydrophobicity of the Rab8a Switch I region was essential for its BAG6-mediated degradation (Fig 4), and thus prevented the excess

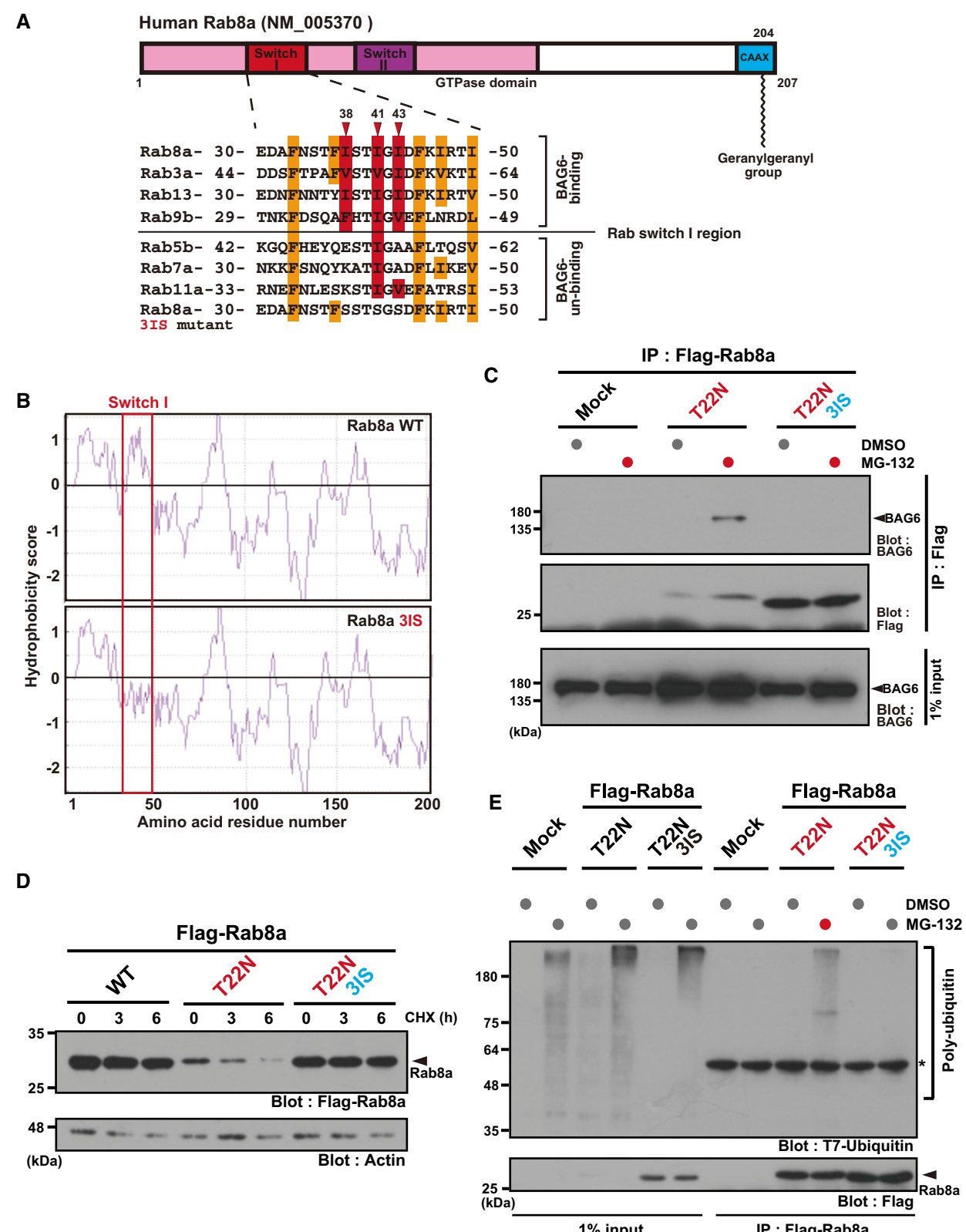

**Figure 4.**

**Figure 4.  The Switch I region of Rab8a is critical for BAG6 recognition and instability.**

A   Schematic representation of the two Switch regions (I and II) within the Rab8a GTPase domain. Numbers denote the corresponding amino acids of human Rab8a (upper panel). Amino acid sequence alignments of the Switch I region of Rab family proteins (lower panel). Three conserved hydrophobic residues (Ile[38], Ile[41], and Ile[43]) in this region are indicated in red, while the other hydrophobic residues are indicated in orange. The three hydrophobic residues were substituted with serine and this construct was designated as the T22N-3IS mutant. Note that Ile[38] and Ile[43] of Rab8a are not conserved in Rab7.
B   Kyte-Doolittle hydrophobicity plots of the complete amino acid sequence of human WT Rab8a and T22N-3IS mutant. The hydrophobicity peak within WT Switch I was abolished in the 3IS mutant (indicated within the red box). The numbers on the horizontal axis denote the corresponding amino acid positions in these proteins.
C   Substitution of the hydrophobic residues of Switch I with hydrophilic residues (T22N 3IS) abolished its binding to BAG6 protein.
D   Rab8a (T22N-3IS) mutant protein was highly stable, while Rab8a (T22N) protein was quite unstable in HeLa cells. Actin was used as a loading control.
E   Rab8a (T22N-3IS) mutant was not subject to polyubiquitin co-precipitation, while Rab8a (T22N) protein was.

Source data are available online for this figure.

accumulation of inactive Rab species during the course of GDP-GTP cycling (Figs 2F, 4D, E, and 5A, C). We favor the idea that BAG6 is an adaptor protein that selectively targets GDP-bound Rab proteins to several BAG6-associated ubiquitin ligases, such as RNF126 and gp78 (Fig EV5) [36,69]. In this context, it should be interesting to note that RNF126 has a role in membrane protein sorting in HeLa cells [81]. Recently, it was reported that some BAG6 client proteins are degraded by the UBXN1-mediated pathway [82]. Therefore, it might be interesting to examine this possibility with Rab8a (T22N) as a substrate.

The mechanism by which some Rab proteins bind BAG6 and others do not is an interesting issue. As shown in Fig 4A, and Appendix Figs S4 and S5, BAG6-phile Rab species tended to possess higher hydrophobicity in the Switch I region, suggesting that this could be one of the factors that determine their affinity with BAG6. This view was further strengthened by our experiments using chimeric Rab8a (T22N) protein with the Rab7-type Switch I sequence (8a-7a chimera, Fig EV3). This experiment revealed that such a chimeric Rab8a (T22N) protein had reduced affinity with BAG6 compared to Rab8a (T22N), suggesting the importance of the Switch I sequence. However, it seems likely that the Switch I sequence is not the sole determinant for BAG6 affinity, since a Rab7 GDP mutant (Rab7a T22N) also showed increased affinity for BAG6, although its affinity was less than that of Rab8a (T22N) (Appendix Fig S7). Collectively, we consider there are at least two critical factors that determine the affinity of Rab protein for BAG6: One is the Switch I sequence (exposed hydrophobicity) and the other is the ratio of the GDP-/GTP-bound forms of respective Rab species, which is greatly influenced by their specific GEF/GAP interactions in the cells. Indeed, our experiments showed that reducing Rab8a GEF (Rabin8/Rab3IP) activity greatly enhanced its affinity for BAG6 (Fig 2F).

An increasing amount of evidence suggests that the excess accumulation of the GDP-bound form of Rab proteins in the cell will lead to the Rab cycle being blocked [72], and cause deleterious effects on vesicular trafficking [15,25,72,73,75–78]. For example, the forced expression of Rab8a (T22N) reduced insulin-stimulated GLUT4 translocation in C2C12 cells [78], modulated Rabin8 distribution and promoted its polarized transport [48], inhibited the transformation of macropinosomes into tubules [15], and inhibited cilia formation [15,65,83]. As Rab8a localizes to RE [84], the misaccumulation of the GDP-bound form of Rab8 might interfere with the retrieval pathways back to the TGN from early endosomes through RE. Similarly, the over-expression of a dominant-negative allele of Rab13 (Rab13 T22N) in MDCK cells disrupted the localization of TGN38/46 in the TGN, and several cargo proteins for surface delivery, such as VSVG, were severely stalled [76,84]. Furthermore, the accumulation of Rab3d dominant-negative mutant proteins, which are deficient in guanine nucleotide-binding, inhibited exocytotic secretion in mouse pancreatic cells [75]. Thus, the inadequacy of Rab (GDP-bound) protein degradation causes the unregulated accumulation of inactive Rab species, possibly not only Rab8a but also other members of the BAG6-phile Rab family, which might impair the nucleotide-binding cycle during the course of endosomal trafficking [24]. Note that the total amounts of Rab8a and Rabin8 were not affected significantly according to Western blot (Fig EV4A) and immunocytochemical (Appendix Fig S8) analyses when BAG6 was depleted. As the GDP-bound form of Rab8a represents the minor population in cells compared with its major GTP-bound form, the degradation of Rab8a-GDP may only have a small effect on the total amount of Rab8a protein.

Since small GTPases constantly switch between different conformational states and might exist frequently in an unstable, aggregation-prone form during the GTP/GDP cycling process [61], the function of GDP-bound/inactive form-specific chaperones should be crucial. Indeed, the REP chaperones newly synthesized Rab GTPases for geranylgeranylation and subsequently escorts isoprenylated Rabs to prevent their cytoplasmic aggregation before membrane anchoring [22]. Another example is that the Rab-binding proteins GDIs recognize Rabs in their GDP-bound form and thereby mediate the dissociation of Rabs from membranes once GTP hydrolysis has been completed and chaperone the geranylgeranylated Rabs in the cytosol [73,85]. Another example is RABIF/MSS4, which forms a complex with nucleotide-free Rab8 and its related proteins [24,86,87]. Structural studies support the assumption that RABIF has a chaperone-like function, and it may act as a suppressor of defective Rab proteins in the cell, thereby preventing the deleterious effects of defective Rabs [25,86,88]. RABIF was proposed to function as a holdase for the nucleotide-free forms of a subset of Rab proteins, such as Rab1b, Rab3a, Rab8a, and Rab10 [24,89]. In mammalian cells lacking RABIF, newly synthesized Rab10 protein is degraded rapidly, likely through the recognition of exposed hydrophobic motifs [27]. In addition to Rab10, Rab8a expression was also diminished in *RABIF*-knockout cells, whereas other Rabs were not significantly affected [27], suggesting that RABIF stabilizes a limited number of Rab family proteins. Although both BAG6 and RABIF seem to recognize exposed hydrophobic residues on Rab proteins, they might possess the opposite (or competing) function for the stability of the inactive forms of Rab proteins. An interesting challenge for the future will be to determine how BAG6, RABIF, and GDIs collaborate with and/or counteract common intrinsically unstable GDP-bound Rab clients. Quantitative and qualitative

                                    

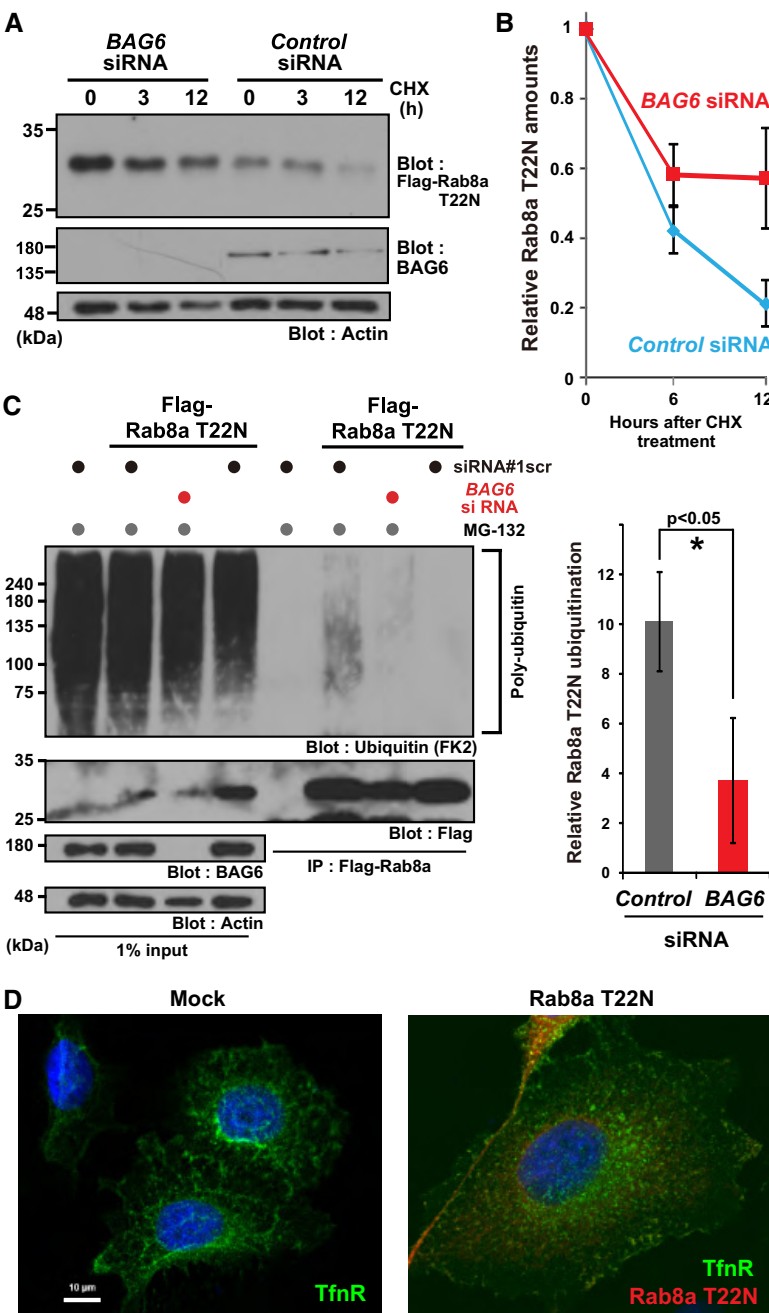

**Figure 5. Endogenous BAG6 is necessary for the elimination of cytosolic Rab8a.**

A   Rab8a (T22N) protein accumulated in BAG6-knockdown cells. HeLa cells were transfected with siRNA duplexes for *BAG6* or control siRNA. At 48 h after siRNA transfection, Flag-tagged-Rab8a (T22N) was expressed in the cells. At 24 h after Rab8a (T22N) transfection, the cells were chased with 50 μg/ml CHX and harvested at the indicated time after CHX addition. Actin was used as a loading control.

B   Anti-Flag blot signals in the control or *BAG6* siRNA-treated cells were quantified, and relative signal intensities after CHX addition were calculated. The value of the Flag-signal at 0 h was defined as 1.0. Note that all signal intensities of the Flag-tag were normalized by that of actin, a loading control, in each sample. The graph represents the mean ± SE calculated from six independent biological replicates. These data were analyzed by Welch's *t*-test.

C   Polyubiquitin modification of Rab8a was abolished in BAG6-knockdown cells. Flag-Rab8a (T22N) immunoprecipitates were blotted with an anti-polyubiquitin antibody (FK2, left panel). As a negative control, siRNA#1scr was used. Anti-polyubiquitin signals co-precipitated with Rab8a (T22N) (a representative example is shown in the left panel) were quantified (right panel). Note that the intensities of the co-precipitated polyubiquitin signal were normalized both by the input ubiquitin-signal and bait Flag-signal. The graph represents the mean ± SD calculated from three independent biological replicates. An asterisk indicates $P < 0.05$ (Student's *t*-test).

D   Defective distribution of the endosomal protein TfnR in HeLa cells with the excess accumulation of the inactive form of Rab8a. Right panel shows a merged image of TfnR staining (shown as green), Rab8a (T22N) staining (magenta), and Hoechst 33342 nuclear staining (blue). Scale bar: 10 μm.

Source data are available online for this figure.

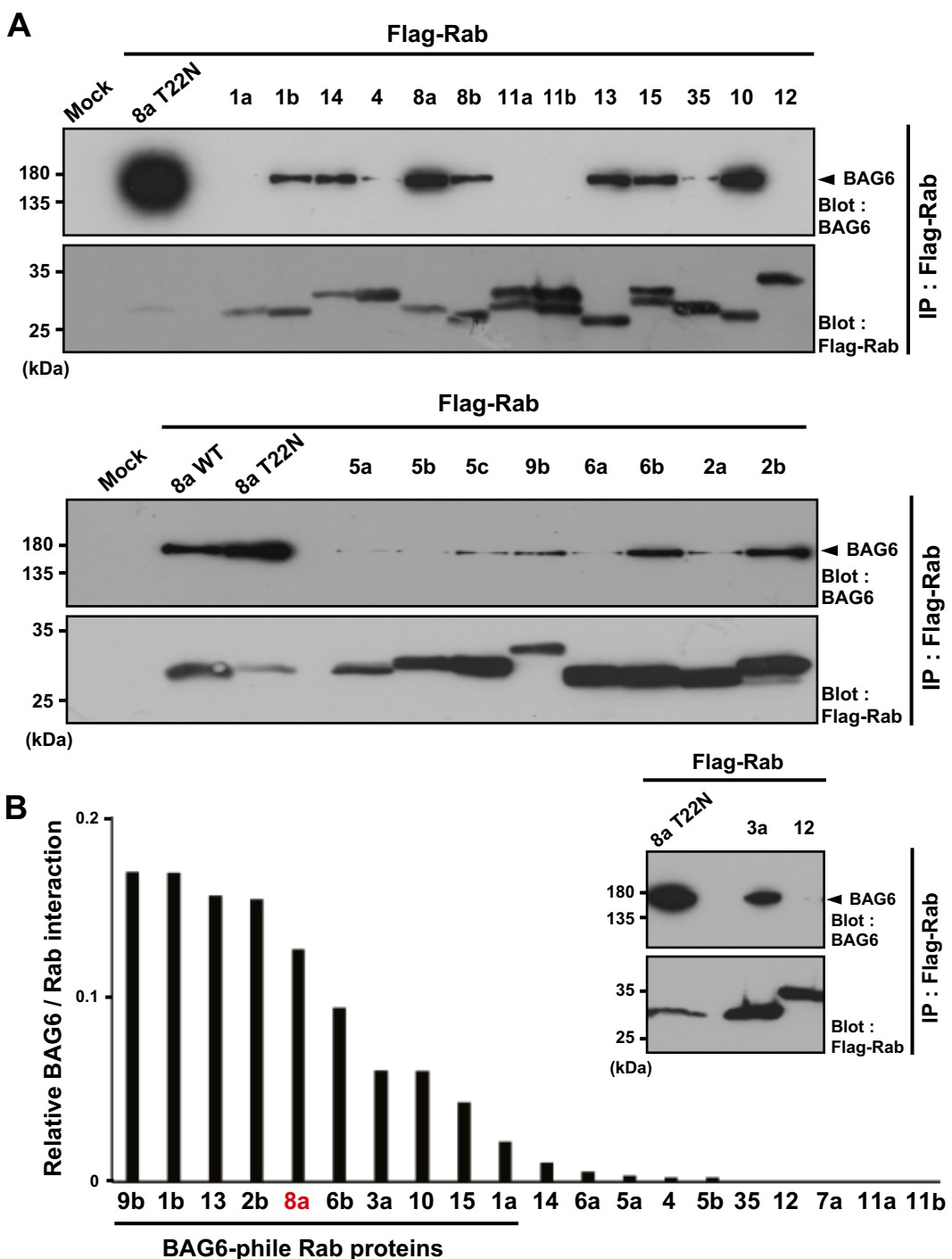

**Figure 6.  BAG6 possesses a distinct preference for multiple, but specific, Rab family proteins.**

A  A series of major Rab family proteins were immunoprecipitated from HeLa cell extracts and then probed with an anti-BAG6 antibody. Co-precipitation of Rab8a (T22N) with BAG6 was used as a positive control for this experiment.

B  Quantification of BAG6 co-precipitation with Rab family proteins. The graph shows the quantities of BAG6 protein in the respective anti-Flag precipitates that were normalized by the amount of Flag-Rab bait proteins. The value of BAG6 co-precipitation with Rab8a (T22N) was defined as 1.0. The quantities of the respective immunosignals were also normalized to the actin signal of each sample and represent the median calculated from at least three independent biological replicates. The number of independent biological replicates was as follows: Rab1a, $n = 4$; Rab1b, $n = 5$; Rab2b, $n = 3$; Rab3a, $n = 4$; Rab4, $n = 4$; Rab5a, $n = 3$; Rab5b, $n = 4$; Rab6a, $n = 5$; Rab6b, $n = 4$; Rab7a, $n = 3$; Rab8a, $n = 14$; Rab9b, $n = 4$; Rab10, $n = 6$; Rab11a, $n = 3$; Rab11b, $n = 3$; Rab12, $n = 2$; Rab13, $n = 6$; Rab14, $n = 3$; Rab15, $n = 5$; Rab35, $n = 3$.

Source data are available online for this figure.

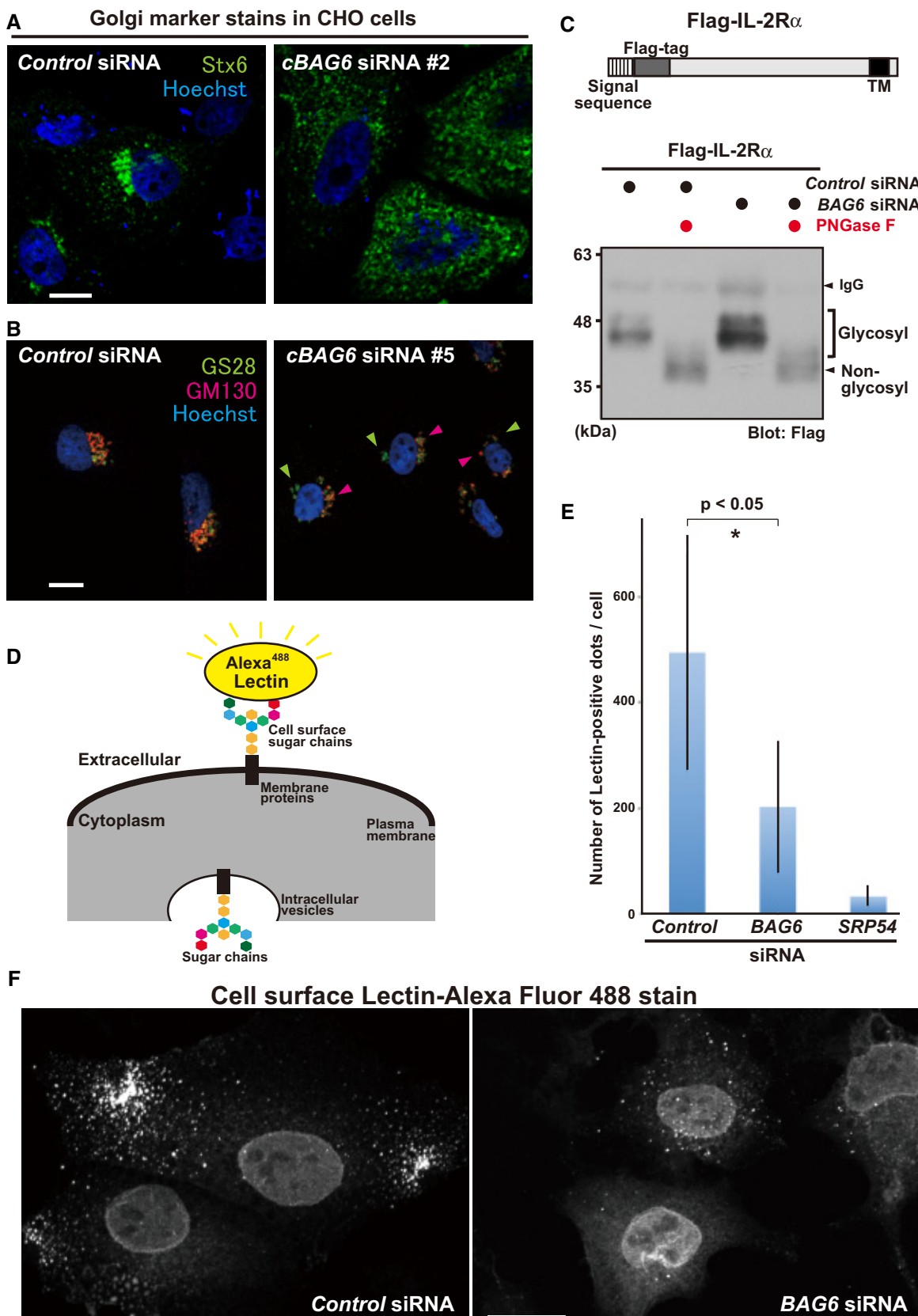

**Figure 7.**

**Figure 7.   Role of BAG6 in the localization of the Golgi apparatus and glycoprotein transport to the plasma membrane.**

A, B    BAG6 knockdown induced the abnormal distribution of Golgi apparatus markers. Representative images of the *trans*-Golgi membrane protein Stx6 (green) in BAG6-suppressed CHO cells with a Chinese hamster-specific siRNA (*cBAG6* siRNA#2). Scale bar: 10 μm (A). Images of the *cis*-Golgi membrane protein GS28 and the *cis*-Golgi matrix protein GM130 in BAG6-suppressed CHO cells with another Chinese hamster-specific siRNA (*cBAG6* siRNA#5). GS28 (green) and GM130 (red) are indicated by arrowheads. Scale bar: 10 μm. (B). Fluorescent signals were detected using a laser scanning confocal microscopy system. Nuclei were stained by Hoechst (blue).

C       Glycosylation of the IL-2Rα transmembrane protein was not reduced by BAG6 knockdown. Flag-tagged WT IL-2Rα protein was expressed in HeLa cells with (+) or without (−) *BAG6* siRNA, and was immunoprecipitated with an anti-Flag antibody. The precipitates were incubated with (+) or without (−) 10 unit of the deglycosylation enzyme PNGase F and subjected to Western blot analysis with an anti-Flag antibody. Low-mobility (indicated as glycosylated) and high-mobility (indicated as non-glycosylated) signals of WT IL-2Rα are indicated.

D–F     Defects in the distribution of cell surface glycoproteins in BAG6-suppressed cells. Representative image from a cell surface glycoprotein quantification assay with Alexa Fluor™ 488-conjugated Lectin GS-II as a probe (D). The graph quantitatively displays the number of fluorescence counts per cell as the mean ± SD calculated from 10 independent biological replicates (E). Lectin GS-II-derived cell surface signals were counted using ImageJ software. SRP54 knockdown was used as a positive control for this experiment (see also Appendix Fig S6). *$P < 0.05$ compared with control knockdown (Student's *t*-test). Cell surface fluorescent signals were detected by confocal microscopy without plasma membrane permeabilization. *BAG6* siRNA down-regulated the cell surface expression of glycoproteins (F).

Source data are available online for this figure.

control of the cytoplasmic pool of Rab family proteins should be a relevant prospect for extensive future investigations, since the regulatory system for the GDP/GTP cycle of small GTPases is intimately linked with various physiological and pathological phenomena, including membrane vesicle trafficking, organelle assembly, and endo/exocytotic events. The findings of the present study will provide important clues to reveal these critical points.

# Materials and Methods

### RNA interference

*BAG6* depletion in human cells was performed as described previously [30] with three independent duplex siRNAs covering the targeted sequences
5′-UUUCUCCAAGAGCAGUUUAtt-3′ (*BAG6* siRNA#1),
5′-CAGAAUGGGUCCCUAUUAUtt-3′ (*BAG6* siRNA#2), and
5′-GAGGAUCAGCGGUUGAUCAtt-3′ (*BAG6* siRNA#3).

*Rab8a* depletion in human cells was performed as described previously [90] with duplex siRNAs covering the targeted sequences
5′-GCAUCAUGCUGGUCUACGAtt-3′ (*Rab8a* siRNA#1),
5′-CACACGUUGUAUAUUCAGAtt-3′ (*Rab8a* siRNA#2), and
5′-CAGCGCGAAGGCCAACAUCAAtt-3′ (*Rab8a* siRNA#3).

Note that the target sequence of *Rab8a* siRNA#1 is conserved in the hamster *Rab8a* gene, and this duplex RNA depleted the corresponding protein in CHO cells efficiently (Fig EV4A).

For *SRP54* depletion in human cells, the duplex siRNA sequence 5′-CACUUAUAGAGAAGUUGAAtt-3′ was synthesized as described previously [42].

*BAG6* depletion in rodent CHO cells was performed with independent duplex siRNAs covering the targeted sequences
5′-GACAUUCAGAGCCAGCGAAtt-3′ (*cBAG6* siRNA#2) and
5′-GACACUUCCUGAAGAGCCAtt-3′ (*cBAG6* siRNA#5).

MISSION siRNA Universal Negative Control 1 (Sigma-Aldrich) was used as a general negative control in every experiment [91]. We also designed scrambled siRNA for *BAG6* siRNA#1 (human) and *cBAG6* siRNA#2 (hamster) as negative controls for BAG6 knockdown experiments in human and hamster cells as follows:
5′-UCUGCAAAUGUCCUUUAAGtt-3′ (*BAG6* siRNA#1scr),
5′-AUGAGCCGUAAAGCGAACCtt-3′ (*cBAG6* siRNA#2 scr-1), and
5′-GUUAACAACCGGGCAGACAtt-3′ (*cBAG6* siRNA#2 scr-2).

Transfections with duplex siRNA were performed using Lipofectamine 2000 (Invitrogen) or Lipofectamine RNAiMAX (Invitrogen), according to the protocols provided by the manufacturer. The efficacy of each siRNA was verified by immunoblot with their specific antibodies listed in the next section.

### Immunological analysis

For immunoprecipitation analysis, HeLa cells were washed with phosphate-buffered saline (PBS) and lysed with immunoprecipitation (IP) buffer containing 50 mM Tris–HCl pH 7.5, 5 mM EDTA, 150 mM NaCl, 10% glycerol, 1% Nonidet P-40, 10 mM N-ethylmaleimide, 20 μM MG-132, and protease inhibitor cocktail (Nacalai tesque). In the immunoprecipitation experiments in Fig 6, 1% Tween-20 was used in the IP buffer as a detergent instead of 1% Nonidet P-40. The lysates were pipetted, centrifuged at $20,630 \times g$ for 10–20 min at 4°C, and mixed with 4–10 μl of anti-Flag M2 affinity gel (Sigma) or anti-T7 tag antibody agarose (Novagen) for 5 min to 2 h at 4°C. After the beads had been washed five times with the IP buffer or chaperone buffer, the immuno-complexes were eluted by SDS sample buffer. For some experiments, we precipitated S-tagged BAG6 from cell lysates. The S-tag is a 15-amino-acid-long peptide (KETAAAKFERQHMDS), and fusion proteins with an N-terminal 2× S-tag can be precipitated efficiently by non-IgG affinity protein (S-protein)-conjugated agarose beads (69704; Merck KGaA, Germany).

For Hot lysis analysis, HeLa cells were washed by PBS and lysed with Hot lysis buffer containing 1% SDS, 50 mM Tris–HCl pH 7.5, 150 mM NaCl, 5 mM EDTA, 20 μM MG-132, 10 mM N-ethylmaleimide, and inhibitor cocktail (Nacalai tesque, Japan). The lysates were heated in 90°C for 15 min. After the lysates were centrifuged at $20,630 \times g$ for 20 min at room temperature, the supernatant was diluted for fourfold with buffer A (containing 1% Triton X-100, 50 mM Tris–HCl pH 7.5, and 150 mM NaCl) and mixed with 6 μl anti-Flag M2 affinity gel for 10 min at 4°C. After the beads had been washed five times with buffer A, the immuno-complexes were eluted by SDS sample buffer.

For Western blot analyses, whole cell lysates and the immuno-precipitates were subjected to SDS–PAGE and transferred onto polyvinylidene fluoride transfer membrane (GE Healthcare, Pall Corporation). The membranes were then immunoblotted with specific antibodies as indicated and then incubated with horseradish

peroxidase-conjugated antibody against mouse or rabbit immunoglobulin (GE Healthcare), followed by detection with ECL Western blotting detection reagents (GE Healthcare) and Clarity™ Western ECL substrate (Bio-Rad).

The following antibodies were used in this study: anti-BAG6 rabbit polyclonal [30], anti-Syntaxin 5 monoclonal [8], anti-Rab8a monoclonal (610844, BD Transduction Laboratories), anti-GS28 monoclonal clone HFD9 (ADI-VAM-PT047, Enzo Life Sciences, Inc.), Alexa Fluor$^R$ 647 anti-GM130 mouse monoclonal (BD biosciences), anti-polyubiquitin FK2 (MBL or Nippon Bio-Test Laboratories Inc.), anti-Flag M2 monoclonal (Sigma), anti-Flag polyclonal (Sigma), anti-T7-tag monoclonal (Novagen), anti-β-actin (Sigma), anti-calnexin polyclonal (Sigma), anti-α-tubulin (TU-02) monoclonal (Santa Cruz Biotechnology, Inc.), anti-α-tubulin (DM1A) monoclonal (Sigma), anti-S peptide (Santa Cruz Biotechnology), and anti-Rabin8/Rab3IP (12321-1-AP, Proteintech for Fig 2F, Homma & Fukuda for Appendix Fig S8B) [92].

## Immunocytochemical observations of Golgi/ER/endosome-localized proteins

For the immunostaining of Flag-TfnR, HeLa cells were fixed by 4% paraformaldehyde-PBS, and permeabilized by 0.1% Triton X-100 solution. For the immunostaining of Ptc1 in HeLa cells, we used pCI-neo-based expression plasmid with largely compromised promoter activity (by deleting promoter region partially) to keep the expression of Flag-tagged Ptc1 protein (Ptc1-Flag) at nearly physiological levels. This was done to preclude the possibility of inappropriate aggregation of this polytopic TMD protein in the cytosol. Transfected cells were grown on microcoverglass (Matsunami, Japan), fixed by incubating in 4% paraformaldehyde for 30 min on ice, and permeabilized with 0.1% Triton X-100 for 3 min at room temperature. For immunocytochemical observations of ER luminal protein calnexin and endosome marker protein Rab7, HeLa cells were fixed with PBS containing 4% paraformaldehyde for 60 min at RT, and cells were then permeabilized with 0.1% Triton X-100 in PBS at 4°C for 5 min. For the immunostaining of GS28, HeLa cells were fixed with PBS containing 4% paraformaldehyde at 4°C for 20 min. After washing cells with PBS, cells were permeabilized with 0.05% (50 μg/ml) digitonin in PBS at 37°C for 5 min. Note that permeabilization with 0.1% Triton X-100 shows different staining pattern of GS28 and makes it difficult to distinguish the effects of BAG6 knockdown compared to the case in control knockdown. For the immunostaining of T7-tagged Stx6, a *trans*-Golgi network marker, CHO cells were grown on microcoverglass (Matsunami, Japan), fixed by incubating in 4% paraformaldehyde for 30 min on ice, and then permeabilized with 0.05% digitonin for 10 min in room temperature.

All cells after fixation and permeabilization were blocked with 3% CS solution in PBS for 30 min at RT, reacted with appropriate primary antibodies as indicated at RT for 60 min or at 4°C for overnight, and were subsequently reacted with secondary antibodies, Alexa Fluor$^R$ 488 goat anti-mouse IgG or Alexa Fluor$^R$ 594 goat anti-rabbit IgG antibodies. To observe the nucleus, cells were treated with 2.5 μg/ml Hoechst 33342. Immunofluorescent images were obtained by Laser scanning confocal microscopy system LSM710 (Carl Zeiss) or by BIOREVO BZ9000 fluorescence microscope (Keyence, Japan).

## Cell surface glycoprotein expression assay

As an assay for glycoprotein transport to the cell surface plasma membrane, a fluorescent conjugate of lectin (Lectin GS-II, Alexa Fluor™ 488 conjugate, L21415; Life Technologies), which is specific for terminal, non-reducing α- or β-linked *N*-acetyl-D-glucosamine residues, was used as a probe. HeLa cells treated with or without BAG6 siRNA for 72 h were fixed with PBS containing 4% paraformaldehyde for 30 min at RT. Alexa Fluor™ 488 conjugate Lectin GS-II was subsequently reacted at room temperature for 1 h or 4°C for 24 h, washing extensively with PBS, and cell surface fluorescent signals were detected by confocal microscopy LSM710 (Carl Zeiss). Note that we did not permeabilized plasma membrane with this assay, since GS-II has been reported to stain the Golgi apparatus as well [93]. ImageJ 1.45s (National Institutes of Health) was used for image processing and quantification of Lectin GS-II-derived cell surface signals.

## Subcellular fractionation

After 72 h of BAG6 siRNA, cells were harvested at $600 \times g$ centrifugation and gently crushed by Dounce-type tissue homogenizer with hypotonic buffer [50 mM HEPES (pH 7.4), 10 mM KCl. 1 mM DTT, protease inhibitor cocktail] on ice. The cell homogenates were centrifuged at $3,000 \times g$ for 2 min and ultra-centrifuged subsequently at $100,000 \times g$ for 20 min at 4°C using Beckman Coulter Optima™ MAX TLS-55 rotor. The resulting $100,000 \times g$ supernatants were used as cytosolic fractions, while the precipitates were used as membrane-containing fractions. After extensive washing the membrane fraction with the hypotonic buffer, the precipitates and the supernatants were dissolved in SDS–PAGE sample buffer, respectively, for Western blot analysis. The successful isolations of membrane-containing and membrane-free fractions were verified by anti-calnexin immunoblots. Tubulin was used as a cytoplasmic marker.

## Glycosyl modification assay for TMD model protein

Flag-tagged IL-2Rα WT protein was expressed in HeLa cells that were treated with (+) or without (−) BAG6 siRNA, and immunoprecipitated with an anti-Flag antibody. The precipitates were incubated with (+) or without (−) five unit of the deglycosylation enzyme PNGase F at 37°C for 2 h *in vitro*, and subjected to Western blot analysis with an anti-Flag antibody [42].

## Statistics

Data are presented as mean ± SD or mean ± SE, and were analyzed using a Student's *t*-test, if not stated otherwise. All analyzed experiments used biological replicates to compute statistical significance. In all statistical analysis, a *P*-value < 0.05 was considered statistically significant.

**Expanded View** for this article is available online.

## Acknowledgements

We thank Prof. Masayuki Komada (Tokyo Inst. Tech.), Prof. Shigeo Murata (Univ. Tokyo), and Dr. Naoto Yokota (Tokyo Metropol. Univ.) for valuable suggestions

and Mr. Kazuyasu Shoji (Tohoku Univ.), Dr. Hiroshi Egawa, Ms. Chizuru Ushio, and Ms. Kotone Izumida (Tokyo Metropol. Univ.) for technical assistance. This work was supported in part by grants from the Ministry of Education, Culture, Science and Technology of Japan (Ubiquitin neo-biology, No. 24112007), the Uehara Memorial Foundation, and Naito Foundation to HK.

## Author contributions

Conceived and designed the experiments: TT, SM, YT, and HK. Performed the experiments: TT, SM, YT, KT, and NS. Providing key materials: KS, SH, NO, and MF. Wrote the paper: TT and HK.

## Conflict of interest

The authors declare that they have no conflict of interest.

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
