## [Review Process File · EMBO Reports]

Cytoplasmic control of Rab family small GTPases through BAG6

Toshiki Takahashi, Setsuya Minami, Yugo Tsuchiya, Kazu Tajima, Natsumi Sakai,
Kei Suga, Shin-ichi Hisanaga, Norihiko Ohbayashi, Mitsunori Fukuda and Hiroyuki Kawahara

Review timeline:	Submission date:	22 July 2018
	Editorial Decision:	14 August 2018
	Revision received:	13 December 2018
	Editorial Decision:	18 January 2019
	Revision received:	25 January 2019
	Accepted:	29 January 2019

Transaction Report:

1st Editorial Decision

14 August 2018

Thank you for the submission of your research manuscript to our journal. We have now received the full set of referee reports that is copied below.

Your manuscript has been evaluated by three referees, who are either experts in membrane trafficking (ref 1, ref 3) or in protein quality control (ref 2). It appears that the evidence that BAG6 regulates Rab8a degradation appears overall solid and requires relatively minor revision as outlined by referee 2. Yet, the opinion of referee 1 and 3 on the trafficking aspect are somewhat divided. Referee 1 considers the data rather preliminary and unclear. Upon further discussion with referee 3, s/he agreed that it was ... "not always clear which trafficking defects relate to the Rab8 interaction specifically" and emphasized again to compare the defects seen with BAG6 and Rab8 knockdown (see also ref3, point 3). Overall, given that all referees recognize that the major finding presented in the paper, i.e., that Rab8 is a client for BAG6, is important and novel, we have decided to invite you to revise your manuscript for EMBO reports.

Please address all referee concerns (as detailed above and in their reports) and take their suggestions on board. Please address all referee concerns in a complete point-by-point response. Acceptance of the manuscript will depend on a positive outcome of a second round of review. It is EMBO reports policy to allow a single round of revision only and acceptance or rejection of the manuscript will therefore depend on the completeness of your responses included in the next, final version of the manuscript.

Revised manuscripts should be submitted within three months of a request for revision; they will otherwise be treated as new submissions. Please contact us if a 3-months time frame is not sufficient for the revisions so that we can discuss the revisions further.

Supplementary/additional data: Please note that you can only submit up to 5 images as Expanded View. Additional Supplementary material should be supplied as a single pdf labeled Appendix. The Appendix includes a table of content on the first page with page numbers, all figures and their legends. Please follow the nomenclature Appendix Figure Sx throughout the text and also label the figures according to this nomenclature. For more details please refer to our guide to authors.

Regarding data quantification, please ensure to specify the name of the statistical test used to generate error bars and P values, the number (n) of independent experiments underlying each data point (not replicate measures of one sample), and the test used to calculate p-values in each figure legend. Discussion of statistical methodology can be reported in the materials and methods section, but figure legends should contain a basic description of n, P and the test applied. Please also include scale bars in all microscopy images.

We now strongly encourage the publication of original source data with the aim of making primary data more accessible and transparent to the reader. The source data will be published in a separate source data file online along with the accepted manuscript and will be linked to the relevant figure. If you would like to use this opportunity, please submit the source data (for example scans of entire gels or blots, data points of graphs in an excel sheet, additional images, etc.) of your key experiments together with the revised manuscript. Please include size markers for scans of entire gels, label the scans with figure and panel number, and send one PDF file per figure.

- a complete author checklist, which you can download from our author guidelines (<http://embor.embopress.org/authorguide#revision>). Please insert page numbers in the checklist to indicate where the requested information can be found.
 - a letter detailing your responses to the referee comments in Word format (.doc)
 - a Microsoft Word file (.doc) of the revised manuscript text
 - editable TIFF or EPS-formatted figure files in high resolution
- (In order to avoid delays later in the publication process please check our figure guidelines before preparing the figures for your manuscript:
http://www.embopress.org/sites/default/files/EMBOPress_Figure_Guidelines_061115.pdf)
- a separate PDF file of any Supplementary information (in its final format)
 - all corresponding authors are required to provide an ORCID ID for their name. Please find instructions on how to link your ORCID ID to your account in our manuscript tracking system in our Author guidelines (<http://embor.embopress.org/authorguide>).

As part of the EMBO publication's Transparent Editorial Process, EMBO reports publishes online a Review Process File to accompany accepted manuscripts. This File will be published in conjunction with your paper and will include the referee reports, your point-by-point response and all pertinent correspondence relating to the manuscript.

I look forward to seeing a revised version of your manuscript when it is ready. Please let me know if you have questions or comments regarding the revision.

REFeree REPORTS

Referee #1:

This manuscript shows very cleanly that a subset of Rab GTPases is chaperoned by BAG6, a protein thought to function is quality control of nascent transmembrane domain proteins. The authors study Rab8A that is known to be difficult to purify in active form from bacteria because it exposes a hydrophobic surface that leads to aggregation. A role for BAG6 is surprising and will be of interest to cell biologists studying membrane trafficking, but this reviewer is not sure that the finding is up

to the higher bar of suitability for EMBO reports. The later figures show abnormal distribution of STX6 and fluorescent lectin staining but the precise basis for any such changes are not shown and remain unclear. That part of the study is not helpful and decreases the overall assessment of the story as it becomes incremental. Finally, the authors speculate that GDI interacts with BAG6 but show no data--this is really not appropriate and requires additional work.

Minor points:

Some language editing would be beneficial

Line 49, SNAREs do not mediate specificity of docking and fusion as certain SNAREs function in multiple steps.

Referee #2:

This study reports on a key role for Bag6 in the degradation of the GDP-form of certain Rab family members, most notably Rab8a. The authors follow up their initial observation that localization of certain proteins in the endocytic pathway seem to be altered in Bag6-depleted cells. This led them to Rab8a, because knockdown of this protein showed similar phenotypes. The authors found that Bag6 physically interacts with Rab8a via its Switch 1 domain that is specifically exposed in the GDP-bound form. This interaction correlates tightly with ubiquitination and degradation, and mutations that perturb the Bag6-Rab8a interaction also lead to stabilization. The author further find that Bag6 interacts with a number of other Rabs and show evidence for multiple types of trafficking defects.

The study is interesting and important because it appears to identify a physiologically important set of substrates for the quality control factor Bag6 and provides mechanistic insight into what is recognized. The evidence for the Bag6-Rab8a interaction, the role of Bag6 in GDP-Rab8a degradation, and the characterization of the region involved in this interaction are all strong. The weak points are that some conclusions are perhaps over-interpreted, and a few of the data may require additional explanations. Overall, the study should be suitable for EMBO Reports with relatively minor revisions as suggested below.

Major points:

1. The authors begin and end their study with observations of altered trafficking and relate these exclusively to Rabs. However, the Bag6 complex is a key component of the tail-anchored insertion pathway that mediates biogenesis of SNAREs. This means that at least some of the trafficking defects might be due to altered SNARE insertion. Thus, the authors should be more cautious in assigning all of the effects to only altered Rab degradation. For example, the first line of the Discussion says "the present study supports the idea that the targets of BAG6 for controlling membrane trafficking events are members of the small GTPase family." This sentence should be revised to say "the present study supports the idea that one set of targets of BAG6..." The authors should mention the possibility that some of the effects seen in Fig. 7 might be due to SNAREs in addition to Rabs, and that this needs to be investigated in future studies.
2. One experimental deficiency of the study is that there is very little analysis of endogenous Rab8a and its relationship to Bag6. The only example I could see is Fig. EV2B, where endogenous Rab8a is shown to be eliminated in the Bag6 knockdown cells. But this is contrary to expectations: if Bag6 is important for degradation of Rab8a-GDP, then either Rab8a levels should not change when Bag6 is knocked down, or possibly increase. I think the authors need to more precisely examine what happens to Rab8a in cells knocked down for Bag6, and if it truly does decrease, they should provide some type of explanation.
3. Related to point (2), it would be more convincing if the authors could show that endogenous Bag6 interacts with endogenous Rab8a. I realise this may not be possible due to inadequate reagents or other technical limitations, but the experiment is worth attempting, and if it does not work, explaining in the paper why.
4. In Figure 3H, the authors state that Q67L is not poly-ubiquitinated. However, the "input" lane shows very low levels of poly-ubiquitin for that sample, so the reduced poly-ubiquitin in the IP sample is not easy to interpret. This technical flaw should be fixed if possible.

Minor points:

1. In the abstract, the authors say that "Bag6 possesses a novel function" but this is not really true. The function being proposed (quality control and degradation) is well established by many papers; it is the substrate that is new. Thus, the better conclusion is to make the claim that Rab family proteins represent a novel set of substrates for the Bag6 quality control pathway.
2. Pg. 4 - "mis-aggregation" should probably just be "aggregation"
3. Pg. 4 - Bag6 is described as a "ubiquitin-like protein" but this is not really accurate. It is better to say Bag6 is a "chaperone/holdase that contains a ubiquitin-like domain"
4. Pg. 6 - in the last sentence of the Introduction, the authors suggest that Bag6 regulates small GTPases to modulate membrane trafficking. This is not really accurate, as there is no evidence of regulation here. The more accurate conclusion is that Bag6 helps to maintain the integrity of membrane trafficking events by limiting excess Rab-GDP accumulation.
5. In Figure 1C, the authors should mention in the text that T22N is expressed at much lower levels than either wild type or Q67L, and that this is due to increased degradation as will be shown later. If you don't make note of this, a reader might be confused.
6. Pg. 11 - in noting that Bag6 has been implicated in protein triage between ER insertion and degradation, the authors cite 8 or 9 different studies. They should strive to be more precise: the role of Bag6 in ER insertion was shown in Mariappan et al., 2010, Nature; the role of Bag6 in degradation of these failed insertion products was shown in Hessa et al., 2011, Nature; and the role of Bag6 in mediating triage was shown in Shao et al., 2017, Science. Many of the other studies are also important for the general idea that Bag6 is involved in protein degradation, but that is more general than what is described in the sentence. In other places as well, it is better to be more precise about citations.
7. Have the authors compared the hydrophobic nature of the Switch 1 regions of the various Rab proteins to see if the ones that interact with Bag6 are more hydrophobic than the ones that do not interact? They should comment on this in the text.
8. Pg. 21 - I think the authors use "alleviates" incorrectly. Perhaps they mean "causes"

Referee #3:

Over the past few years BAG6 has emerged as a new protein chaperone, diverting misfolded and misassembled proteins from the cytosol to the proteasome for degradation. To date, its recognised targets are limited to transmembrane domain proteins and hydrophobic polypeptides, however in this study the authors expand the repertoire of known BAG6 clients to include a subset of Rab GTPases. More specifically, the authors demonstrate that BAG6 binds Rab8a-GDP, but not Rab8a-GTP, to promote its ubiquitin-mediated proteasomal degradation and prevent aggregation in the cytosol after release from membranes. In the absence of BAG6, Rab8a membrane trafficking events become dysregulated and protein sorting to the plasma membrane, as well as protein glycosylation, is defective.

These findings, which are well presented, are novel and of interest to the molecular and cell biology communities. The manuscript is well written with a clear rationale presented for the experiments performed and overall the biochemistry is of a high standard. However, in order to strengthen the conclusions drawn and the clarity of the manuscript the following revisions are suggested to address some minor criticisms prior to publication.

Technical criticisms:

1. BAG6 is argued to prevent aggregation of Rab8a-GDP in the cytosol however this study provides no evidence that aggregation occurs. It is therefore important to show localisation of Rab8a and/or

rabin8 in BAG6 KD cells to see whether its distribution is affected and not just protein levels. Since other proteins are mislocalised in these cells it should be possible to see some disruption.

2. Fig EV1A is missing a positive control for polyubiquitin stain to show that the antibody is working and what would be expected should polyubiquitination occur.
3. Rab8a KD is shown to cause similar defects in TfnR localisation to BAG6 KD. It would be interesting to see if Rab8 knockdown induced the same abnormal distribution of Golgi apparatus markers, Ptch1 and Rab7 too to clarify which of the presented phenotypes relate to Rab8 and which to the other Rabs. Perhaps the authors could also check if double depletion of Bag6 and Rab8 could rescue some of the observed membrane sorting defects.
4. Authors show in Fig.3 that the T22N mutant, but not active Rab8a, is ubiquitinated. Recently it has been described that BAG6 dependent ubiquitylation process is mediated by VCP-UBXN1 complex (Ganji et al.2018). It would be interesting to address if Rab8a ubiquitylation is also dependent on this complex or if Bag6 acts independently in this case. At the very least perhaps it is worth speculating on and including the reference.

Manuscript criticisms:

1. Fig1 and EV1 please make the images brighter. The DAPI staining is missing or very dim in Fig1A.
2. In all the figures with IPs BAG6 is annotated as S-BAG6. Please specify the meaning of s-BAG6 in the method section or figure legend.
3. The schematic mentioned in the figure legend for BAG6 mutants is missing (Fig3D). Please add this to the figure.
4. One of the most interesting pieces of data is that Bag6 specifically binds to the hydrophobic region present in the Switch I of the GTPase domain (Fig.4). This region is formed in Rab8a by three hydrophobic amino acids. Interestingly these hydrophobic amino acids are absent in Rab7 which does not interact with BAG6 - it would be informative to see the alignment of all the Rab proteins in Fig.6.
5. Linked to the above point, discussion about why some Rabs bind BAG6 and some do not would be useful and interesting. For example, is it the structure of the switch I region or is it Rabs involved in particular pathways/organelles or is it not predicable.
6. Fig. 5B. Quantification of band density at all timepoints would be more useful to see the extent of degradation over time.
7. Fig.6 and the discussion of Fig.6 is not systematic in how it deals with the Rab proteins and there are inconsistencies. For example many of the Rab proteins shown on the blot are then not quantified on the graph ie 35, 6a, a, 5b. Also for some Rabs, the blots presented are not representative of the quantification, for example, a large amount of Rab10 and 1b seems to bind to Bag6 in the blot but not the graph. This may be an artefact of the normalisation and this needs to be looked at more carefully.

1st Revision - authors' response

13 December 2018

Response to Referee #1:

First of all, we would like to express our sincere gratitude to Referee #1 for providing us with a number of insightful and constructive suggestions that have improved our manuscript greatly. We were pleased to read the Referee#1's comments that our manuscript showed very clearly that a subset of Rab GTPases are chaperoned by BAG6, and that a role for BAG6 is surprising and will be of interest to cell biologists studying membrane trafficking. In the revised version, we have addressed all of the comments of Referee #1 with additional experiments as follows. We hope that our manuscript is now acceptable for publication in *EMBO Reports*.

1.
Referee 1 emphasized to compare the defects seen with BAG6 and Rab8 knockdown.

At the head of Editor's letter, Referee #1 pointed out that we should compare the defects seen with BAG6 and Rab8 knockdown. We completely agree with the Referee #1's suggestion. In accordance with this important advice, we examined whether Rab8a knockdown induced the abnormal distribution of the endosomal protein Ptc1 in HeLa cells, and the *trans*-Golgi apparatus marker Stx6 and *cis*-Golgi markers GS28 and GM130 in CHO cells. As a result, we found that Rab8a depletion caused defects in the localization of all of these organelle-associated marker proteins. For example, the distribution of Ptc1 protein was altered greatly to perinuclear compartments and reduced cytoplasmic vesicular dots in Rab8a-depleted cells (Fig EV1B), an almost identical phenotype to that observed in BAG6-depleted cells (Fig 1B and Fig EV1A). The abnormal distribution of Stx6 was observed in Rab8a-depleted CHO cells, which was nearly indistinguishable from the case in BAG6-depleted cells (Fig EV4B). GS28 and GM130 signals were also dispersed throughout the perinuclear region of the cytoplasm, an overlapping phenotype to that observed in BAG6-depleted cells (Fig EV4C). As we had demonstrated in Fig 1A, the distribution of the recycling endosomal protein TfnR was also affected similarly by either BAG6 or Rab8a depletion. All of these observations support our original view that the function of BAG6 is closely related to that of Rab8a. Accordingly, we have added the new Figures EV1B, EV4B, and EV4C and relevant sentences to the Results and Figure Legend sections. Thank you very much for providing us with an opportunity to strengthen our results greatly.

p.6, line 118 -120.

"The defective distribution of Ptc1 was also observed following Rab8a depletion (Fig EV1B), which was a similar phenotype to that observed in BAG6-depleted cells (Fig 1B and Fig EV1A)"

p.16, line 315-317.

"The abnormally dispersed distribution of Stx6 was also observed in Rab8a-depleted CHO cells, which was nearly indistinguishable to the case in BAG6-depleted cells (Fig EV4B)."

p.17, line 322-325.

"In BAG6 knockdown cells, however, their signals were separated with less co-localization throughout the perinuclear region of the cytoplasm (Fig 7B, right panel; compare the red and green signals), a similar phenotype to that observed in Rab8a-depleted cells (Fig EV4C)."

p.52, line 1031-1032.

"**(B)** Immunostaining of Ptc1 (green) in HeLa cells that were treated with or without *Rab8a* siRNA. See also Appendix Fig S1D. Scale bar: 10 μ m."

Appendix Figure legend; p.2, line 33-35.

"**(D)** Immunostaining of the Ptc1 (green) in HeLa cells that were treated with *Rab8a* siRNA (left), *BAG6* siRNA (center) and their combination (right). The positions of the nucleus are shown by Hoechst staining (blue). See also Fig 1B and Fig EV1B. Scale bars: 10 μ m."

2-1.

The later figures show abnormal distribution of STX6 and fluorescent lectin staining but the precise basis for any such changes are not shown and remain unclear. That part of the study is not helpful and decreases the overall assessment of the story as it becomes incremental.

As we explained in the previous section, we showed that the abnormal distribution of the *trans*-Golgi apparatus marker Stx6 in Rab8a-depleted CHO cells, which was nearly indistinguishable from the phenotype observed in BAG6-depleted cells (Fig EV4B), partly supporting the molecular basis for such an abnormality. In addition, the effects of *BAG6* siRNAs on organelle markers were found to be conserved in different species, namely, human (HeLa) and hamster (CHO) cells, with their respective independent duplex siRNA sequences, greatly reducing the potential risk for any non-specific (off-target) effect of our knock down experiments. We believe that this should be

another indicator of the importance of the study of the distribution of Stx6.

p.17, line 327-330

“In addition, these observations indicate that the effects of *BAG6* siRNAs on organelles were conserved in different species, namely, humans (Figs 1 and EV1A) and hamsters (Figs 7A, B, and EV4), with their respective unique double stranded RNA sequences (see Materials and Methods).”

p.24, line 479-p.25, line 482

“*BAG6* depletion in rodent CHO cells was performed with independent duplex siRNAs covering the targeted sequences;

5'- GACAUUCAGAGCCAGCGAAtt -3' (c*BAG6* siRNA#2)

5'- GACACUUCUGAAGAGCCAtt -3' (c*BAG6* siRNA#5)”

p.54, line 1074- p.55, line 1083

“(B) Comparison of the defects observed in the distribution of Stx6 (green) with siRNAs (c*BAG6* siRNA#5 and *Rab8a* siRNA#1, respectively) in CHO cells. Note that both c*BAG6* siRNA#5 and #2 are Chinese hamster-specific, while the target sequence of *Rab8a* siRNA#1 is identical between humans and hamsters. An asterisk indicates a non-specific signal. Scale bar: 10 μ m. (C) Comparison of the defects observed in the distribution of the ER-Golgi SNARE protein GS28 (green) and the *cis*-Golgi marker GM130 (red) with siRNAs (c*BAG6* siRNA#2 or *RAB8a* siRNA#1 and their combination) in CHO cells. GS28 and GM130 signals were dispersed throughout the perinuclear region of the cytoplasm in *Rab8a* knockdown cells, a similar phenotype to that observed in *BAG6*-depleted cells. Scale bar: 10 μ m.”

2-2.

For the fluorescent lectin staining experiments, we do not have adequate evidence about the precise basis for such changes at this moment. Nevertheless, this fact suggests a role for *BAG6* in glycoprotein sorting from the ER/Golgi to the plasma membrane, we think this finding is worth noting in this study. As a next step, we will try to expand this finding, but it will take a long time before we can complete such mechanistic experiments. Since we wish to publish our current findings in a timely fashion in *EMBO Reports*, we sincerely request the reviewer's understanding regarding this point.

3.

The authors speculate that GDI interacts with BAG6 but show no data--this is really not appropriate and requires additional work.

First of all, we would like to apologize for not showing our GDI data, despite mentioning them in the Discussion section. We hesitate to include our GDI data because our analysis of these data is too preliminary for publication. Here, we disclose our GDI data to the Referees as “Data for Referee #1” as shown below. We have already confirmed that this observation is reproducible.

Data for Referee #1:

GDI1 and GDI2 associate with *BAG6*.

Flag-tagged GDI1 and GDI2 were expressed in HeLa cells with S-tagged BAG6, and Flag-precipitates were blotted with an anti-BAG6 antibody. Flag-GDI1 (and to lesser extent, GDI2) was co-precipitated with BAG6 exclusively in the presence of MG-132.

Unfortunately, we do not know the exact physiological significance of this preliminary observation at this moment. We completely agree with Referee #1's comment that it was not appropriate to mention this without showing evidence and the underlying rationale. Accordingly, we decided to remove the description of the BAG6-GDI interaction from the revised Discussion. We sincerely apologize for this alteration. We will try to expand this finding in future experiments, but it will take a long time before we can complete this project, and we sincerely request the reviewer's understanding regarding this point.

4.

Some language editing would be beneficial.

Referee #1 pointed out that the English language of the previous manuscript should be edited. We would like to apologize about this point and the entire manuscript has been scrutinized and corrected by a professional native-speaking English scientist. We hope that the English in the manuscript is now acceptable for publication.

5.

Line 49, SNAREs do not mediate specificity of docking and fusion as certain SNAREs function in multiple steps.

According to this kind suggestion, we have deleted the words "specificity and" as follow:

p.3, line 47.

"which mediate vesicle docking and fusion [8,9],"

We believe that these changes have improved the quality of our data greatly and that our manuscript is now acceptable for publication in *EMBO Reports*.

Thank you very much for your kind support and excellent conceptual suggestions.

Responses to Referee #2

We would like to express our sincere gratitude to Referee #2 for providing us with a number of constructive and important suggestions. We were very happy to read the Referee #2's comments that our study is interesting, important, and that the evidence is all strong. We are also grateful to Referee #2 for kindly recommending our manuscript with the comment that the overall study should be suitable for publication in *EMBO Reports*. In the revised version, we have addressed all of the comments by Referee #2 as follows. We hope that our manuscript is now acceptable for publication in *EMBO Reports*.

At the head of Referee #2's letter, Referee #2 pointed out that the weak points are that some conclusions are over-interpreted, and some of the data may require additional explanations. With this advice, we have changed the relevant parts of the revised manuscript as follows.

1-1.

Since the Bag6 complex is a key component of the tail-anchored insertion pathway that mediates biogenesis of SNAREs. This means that at least some of the trafficking defects might be due to altered SNARE insertion. Thus, the authors should be more cautious in assigning all of the effects to only altered Rab degradation.

The authors should mention the possibility that some of the effects seen in Fig. 7 might be due to SNAREs in addition to Rabs, and that this needs to be investigated in future studies.

We agree with the comments of Referee #2 completely. At the initial stage of this work, we focused on the effects of BAG6 on the biogenesis of several SNARE components, such as GS28, since these proteins belong to the tail-anchored protein family, as suggested by Referee #2. However, our preliminary study failed to show that the membrane incorporation of these tail-anchored proteins was affected significantly under BAG6 knockdown conditions (please see Data for Referee #2), probably due to the redundancy of an alternative tail-anchored protein insertion pathway, as suggested recently (Shao et al, 2017; Casson et al, 2017). This is the reason why we did not mention the possible effects on the altered SNARE insertion in the original manuscript.

Data for Referee #2:

BAG6 knockdown induces no apparent defects in the membrane assembly of tail-anchored protein GS28.

Whole cell lysates from BAG6-depleted cells (*BAG6* siRNA) and control cells (Control siRNA) were fractionated into cytosolic and insoluble membrane fractions. The results showed that a tail-anchored protein GS28 was detected exclusively in the membrane fraction, even under the BAG6-depleted condition. Note that all cells and extracts were treated with MG-132 (and other protease inhibitors) to prevent the degradation of defective tail-anchored proteins. We used calnexin as an ER marker and tubulin as a cytoplasmic marker.

That being said, we cannot exclude the possibility of defects in the membrane insertion of some SNARE proteins in BAG6-suppressed cells. Accordingly, we cited important references (Shao et al, 2017; Casson et al, 2017) and discussed the possibility that some of the trafficking defects might be due to altered SNARE insertion in the Discussion section, as kindly suggested by Referee #2. Furthermore, we mentioned the possibility that some of the effects seen in Fig 7 might be due to SNAREs in addition to Rabs, and that this needs to be investigated in future studies. We greatly appreciate Referee #2 for providing us an opportunity to discuss this point.

“p.19, line 363-367.

“Since the BAG6 complex is also known as a key component of the TA protein insertion pathway, which mediates the biogenesis of SNARE proteins [38,56,68,80], this suggests that some of the observed trafficking defects might be due to SNAREs in addition to Rabs. Such a possibility obviously needs to be investigated in future studies.”

p.40, line 791-792

“68. Shao S, Rodrigo-Brenni MC, Kivlen MH, Hegde RS (2017) Mechanistic basis for a molecular triage reaction. *Science* **355**: 298-302”

p.42, line 823-825

“80. Casson J, McKenna M, Haßdenteufel S, Aviram N, Zimmerman R, High S (2017) Multiple pathways facilitate the biogenesis of mammalian tail-anchored proteins. *J*

Cell Sci **130**: 3851-3861.”

1-2.

For example, the first line of the Discussion says "the present study supports the idea that the targets of BAG6 for controlling membrane trafficking events are members of the small GTPase family." This sentence should be revised to say "the present study supports the idea that one set of targets of BAG6..."

We agree with Referee #2's suggestion. According to this comment, we have corrected our sentence exactly as Referee #2 kindly suggested.

p.19, line 362-363.

“This study supports the idea that members of the small GTPase Rab family are one set of targets of BAG6 for controlling membrane trafficking events.”

2-1.

One experimental deficiency of the study is that there is very little analysis of endogenous Rab8a and its relationship to Bag6.

According to this comment, we have added our analysis of endogenous Rab8a protein and its relationship to BAG6 in Fig EV2A, as discussed in detail in following section 3.

2-2.

The only example I could see is Fig. EV2B, where endogenous Rab8a is shown to be eliminated in the Bag6 knockdown cells. But this is contrary to expectations: if Bag6 is important for degradation of Rab8a-GDP, then either Rab8a levels should not change when Bag6 is knocked down, or possibly increase. I think the authors need to more precisely examine what happens to Rab8a in cells knocked down for Bag6, and if it truly does decrease, they should provide some type of explanation.

We completely agree with Referee #2. According to this comment from Referee #2, we have repeated this experiment, and confirmed our previous findings that Ptc1 proteins were reproducibly increased both under BAG6 and Rab8a knockdown conditions. Since we think this result is so important, we decided to move it from the previous Fig EV2B to the new Fig 1C. Thus, we would like to emphasize that our main conclusion for this experiment was confirmed.

The problem in our previous Fig EV2B was that endogenous Rab8a, shown as a Rab8a knockdown control, seemed to be eliminated in the BAG6 knockdown cells for an unknown reason, but this was contrary to expectations, as pointed out by Referee #2. Accordingly, we examined in detail what happened to endogenous Rab8a in BAG6 knockdown cells. Our repeated experiments showed that the amount of endogenous Rab8a was not decreased in BAG6-suppressed HeLa cells (please see the revised Figs 1C and EV4A). We further confirmed that the case was similar in CHO cells. When we depleted endogenous BAG6 from CHO cells with independent duplex siRNA, the signal intensity of endogenous Rab8a was not affected, neither in western blot analysis (Fig EV4A) nor in immunocytochemical analysis (Appendix Fig S8C). Since it is likely that the GDP-bound form of Rab8a protein is a minor population compared with its GTP-bound form, the degradation of Rab8a-GDP would have only a minor impact on the total amount of Rab8a protein. Therefore, as suggested by Referee#2, the total amount of Rab8a did not change (or increased slightly) when BAG6 was knocked down. Our experiments support this conclusion directly.

p.21, line 418- p.22, line 423

“Note that the total amounts of Rab8a and Rabin8 were not affected significantly according to western blot (Fig EV4A) and immunocytochemical (Appendix Fig S8) analyses when BAG6 was depleted. As the GDP-bound form of Rab8a represents the minor population in cells compared with its major GTP-bound form, the degradation of Rab8a-GDP may only have a small effect on the total amount of Rab8a protein.”

p.44, line 872-874

“(C, D) Knockdown of Rab8a (with *Rab8a* siRNA#1, #2, and #3) or BAG6 (with *BAG6*

siRNA#1) stimulated the accumulation and stabilization of Ptc1 protein in HEK293 cells. See also Appendix Fig S1B.”

Appendix Figure Legends: p.5, line 94-100.

“Appendix Figure S8. In relation to Fig EV4A, no detectable abnormality of Rab8a and Rabin8 localizations in BAG6-suppressed cells. Immunostaining of the endogenous Rab8a (A, C, shown as green) in HeLa (A) and CHO (C) cells. Immunostaining of the endogenous Rabin8 in HeLa cell (B, shown as green). The positions of the nucleus are shown by Hoechst staining (blue). All cells were transfected with a series of siRNA constructs as indicated. Scale bars: 10 μ m.”

3.

Related to point (2), it would be more convincing if the authors could show that endogenous Bag6 interacts with endogenous Rab8a. I realize this may not be possible due to inadequate reagents or other technical limitations, but the experiment is worth attempting, and if it does not work, explaining in the paper why.

According to this convincing suggestion, we tried to examine whether endogenous Rab8a interacts with BAG6. As shown in the new Figure EV2A, we showed that endogenous Rab8a protein can be co-precipitated with an S-tagged BAG6 fragment from cell lysates. Our difficulty was to precipitate endogenous BAG6 due to technical limitations, as mentioned in Referee #2’s comment. Accordingly, we have added relevant sentences and a new Figure EV2A as follows.

p.8, line 155-156.

“We also confirmed that endogenous Rab8a protein was associated with BAG6 (Fig EV2A).”

p.45, line 896-897

“(E) A series of Flag-Rab8a mutants were immunoprecipitated and quantified the amount of endogenous BAG6 that were coprecipitated with Flag-Rab8a.”

p.52, line 1039-1041.

“(A) Endogenous Rab8a protein was co-precipitated with S-tagged BAG6 N465 from HeLa cell lysate in the presence of MG-132.”

4.

In Figure 3H, the authors state that Q67L is not poly-ubiquitinated. However, the "input" lane shows very low levels of poly-ubiquitin for that sample, so the reduced poly-ubiquitin in the IP sample is not easy to interpret. This technical flaw should be fixed if possible.

According to this comment, we have repeated the immunoprecipitation experiments, and compared polyubiquitin co-precipitations between WT and Q67L Rab8a. As shown in Appendix Fig S2B, we confirmed that the Q67L mutant was subjected to less or a similar amount of polyubiquitination compared with the case in WT Rab8a. Accordingly, we added Appendix Fig S2B and modified the relevant sentences in the Results sections as follows:

p.11, line 221-p.12, line 223.

“Highly stable Rab8a (Q67L) and WT Rab8a (a mixture of predominant GTP-bound and minor GDP-bound forms) were subjected to low-level polyubiquitination (Fig 3H and Appendix Fig S2B) in contrast to the case for Rab8a (T22N).”

Appendix Figure Legends; p.2, line 44-45.

“(B) Highly stable Rab8a (Q67L) was subjected to polyubiquitin modification only at a low level even in the presence of the proteasome inhibitor (see also Fig 3H).”

5.

In the abstract, the authors say that "Bag6 posses a novel function" but this is not really true. The function being proposed (quality control and degradation) is

well established by many papers; it is the substrate that is new. Thus, the better conclusion is to make the claim that Rab family proteins represent a novel set of substrates for the Bag6 quality control pathway.

According to this comment, we have corrected our sentence exactly as Referee #2 kindly suggested.

p.2, line 33-36.

“From these observations, we suggest that 355 Rab proteins represent a novel set of substrates for BAG6, and the BAG6-mediated pathway is associated with the regulation of membrane vesicle trafficking events in mammalian cells.”

6.

Pg. 4 - "mis-aggregation" should probably just be "aggregation"

We have corrected our sentence exactly as Referee #2 kindly suggested as follow.

p.4, line 66-67.

“have been reported to help prevent the aggregation of Rab proteins”

7.

Pg. 4 - Bag6 is described as an "ubiquitin-like protein" but this is not really accurate. It is better to say Bag6 is a "chaperone/holdase that contains a ubiquitin-like domain"

We have corrected our sentence exactly as Referee #2 kindly suggested.

p.4, line 69-70.

“BAG6 (also called BAT3 or Scythe) is a chaperone/holdase that contains a ubiquitin-like domain and interacts with aggregation-prone hydrophobic polypeptides”

8.

Pg. 6 - in the last sentence of the Introduction, the authors suggest that Bag6 regulates small GTPases to modulate membrane trafficking. This is not really accurate, as there is no evidence of regulation here. The more accurate conclusion is that Bag6 helps to maintain the integrity of membrane trafficking events by limiting excess Rab-GDP accumulation.

According to the kind suggestion from Referee #2, we have corrected our sentence as suggested:

p.5, line 93-96.

“From these observations, we suggest that BAG6 possesses an unexpected but critical function in maintaining the integrity of membrane trafficking events by limiting the excess accumulation of GDP-bound forms of Rab small GTPases.”

9.

In Figure 2C, the authors should mention in the text that T22N is expressed at much lower levels than either wild type or Q67L, and that this is due to increased degradation as will be shown later. If you don't make note of this, a reader might be confused.

We are grateful to Referee #2 for bringing this to our attention, and we mentioned this fact in the text as follow:

p.45, line 887-889.

“Note that the T22N mutant protein was expressed at lower levels than either WT or Q67L, and that this was partly due to increased degradation, as will be shown later.”

10-1.

Pg. 11 - in noting that Bag6 has been implicated in protein triage between ER insertion and degradation, the authors cite 8 or 9 different studies. They should strive to be more precise: the role of Bag6 in ER insertion was shown in Mariappan et al., 2010, Nature; the role of Bag6 in degradation of these failed insertion products was shown in Hessa et al., 2011, Nature; and the role of Bag6 in mediating triage was shown in Shao et al., 2017, Science. Many of the other

studies are also important for the general idea that Bag6 is involved in protein degradation, but that is more general than what is described in the sentence.
Thank you very much for your suggestion. According to the comments from Reviewer #2, we have corrected our citation as Referee #2 kindly suggested.

p.9, line 179- p.10, line 182.

“BAG6 has been reported to be a cytoplasmic triage factor [68], and is critical for both tail-anchored (TA) protein insertion [38,56] and the degradation of failed insertion products [31]. BAG6 possess a strong preference for binding to the exposed hydrophobic residues of membrane proteins in the cytosol [31,35,42,43.]”

p.37, line 727-728.

“43. Juszkiwicz S, Hegde RS. (2018) Quality control of orphaned proteins. *Mol Cell* **71**: 443-457”

p.40, line 791-792.

“68. Shao S, Rodrigo-Brenni MC, Kivlen MH, Hegde RS (2017) Mechanistic basis for a molecular triage reaction. *Science* **355**: 298-302”

10-2.

In other places as well, it is better to be more precise about citations.

According to the suggestion from Reviewer #2, we have corrected citations more precisely as follows:

p.4, line 69-79.

“BAG6 (also called BAT3 or Scythe) is a chaperone/holdase that contains a ubiquitin-like domain and interacts with aggregation-prone hydrophobic polypeptides and escorts them to the degradation machinery [28-35]. BAG6 possesses an intrinsic affinity for the hydrophobic residues of client proteins [28,30-32,36,37] and captures newly synthesized polypeptides by means of their exposed hydrophobic patches concomitant with or after their release from the ribosome [30,31,38], which improves their solubilization, assembly, and/or degradation efficiency [31,32,35,36,39]. BAG6 also captures and shields the exposed transmembrane domain (TMD) of newly synthesized TMD proteins in the cytosol for subsequent degradation if their proper biogenesis has failed [31,35,39-43]. Thus, a series of studies have shown crucial roles for BAG6 in the quality control of newly synthesized TMD proteins [31-33,35,40,41,43,44].”

11.

Have the authors compared the hydrophobic nature of the Switch I regions of the various Rab proteins to see if the ones that interact with Bag6 are more hydrophobic than the ones that do not interact? They should comment on this in the text.

According to this insightful suggestion, we have provided amino acid sequence alignment of the Switch I region of all Rab proteins listed in Fig 6 (Appendix Fig S4). In addition to this information, we examined the hydrophobicity profile of Rab family proteins with a Kyte-Doolittle hydrophobicity plot (provided as Appendix Fig S5). As you will see, BAG6-phile Rab species (such as Rab8a, 3a, 2b and 6b) tended to possess a higher hydrophobicity peak in the Switch I region (boxed as a red rectangle in Appendix Fig S5) compared with Rab proteins with weak BAG6 interactions (such as Rab7a, 12, 11b, and 4a). Accordingly, we provided two new figures as Appendix Figs S4 and S5, and relevant sentences in the Appendix Figure Legends as follows.

p.16, line 301-304.

“Amino acid sequence alignments of the Switch I regions (Appendix Fig S4) and Kyte-Doolittle hydrophobicity plots (Appendix Fig S5) suggest that BAG6-phile Rab species tended to possess a higher hydrophobicity peak in the Switch I region compared with Rab proteins with weak BAG6 interactions.”

Appendix Figure Legend, p.3, line 63-p.4, line 69.

“Appendix Figure S4. In relation to Figs 4A and 6B, amino acid sequence alignments of

the Switch I region of the Rab proteins. The distribution of the hydrophobic residues in the Switch I region is indicated in color. The hydrophobic residues that are frequently found in BAG6-phile Rab proteins are shown in red (indicated by red arrowheads), while the other hydrophobic residues are shown in orange.”

Appendix Figure Legend, p.4, line 71-76.

“Appendix Figure S5. In relation to Appendix Fig S4, Kyte-Doolittle hydrophobicity plots of BAG6-phile Rab proteins and that with weak BAG6 interaction. The hydrophobicity peak within Switch I region of respective Rab proteins are indicated as red rectangle. The horizontal axis numbers denote the corresponding amino acid positions in these proteins.”

12.

Pg. 21 - I think the authors use "alleviates" incorrectly. Perhaps they mean "causes"

Thank you very much for bringing these errors to our attention. We have corrected our sentence exactly as Referee #2 kindly suggested.

p.21, line 415-416.

“Thus, the inadequacy of Rab (GDP-bound) protein degradation causes the unregulated accumulation of inactive Rab species.”

We hope that the changes have further improved our manuscript greatly and that our manuscript is now acceptable for publication in the *EMBO Reports*.

Thank you very much for your kind support.

Responses to Referee #3

We would like to express our sincere gratitude to Referee #3 for providing us with the positive comment that our findings are novel and of interest to the molecular and cell biology communities. We were also very happy to read that our manuscript was well written with a clear rationale presented for the 520 experiments performed and overall the biochemistry was of a high standard. We are grateful for Referee #3's suggestion to strengthen the conclusions drawn and the clarity of the manuscript. In the revised version for *EMBO Reports*, we have addressed all of the comments from Referee #3 as follows.

Technical criticisms:

1.

BAG6 is argued to prevent aggregation of Rab8a-GDP in the cytosol however this study provides no evidence that aggregation occurs. It is therefore important to show localisation of Rab8a and/or rabin8 in BAG6 KD cells to see whether its distribution is affected and not just protein levels. Since other proteins are mislocalised in these cells it should be possible to see some disruption.

According to this insightful suggestion, we tried to examine the localization of endogenous Rab8a in BAG6 knockdown cells to see whether its distribution is affected. Unfortunately, we did not detect such an abnormality, including aggregation, of endogenous Rab8a protein in BAG6-suppressed HeLa (Appendix Fig S8A) and CHO (Appendix Fig S8C) cells. We suspect this was partly because the majority of the signal detected by the anti-Rab8a antibody is derived from the stable GTP-bound form of Rab8a (please note that the GTP-bound form is not a target of BAG6), while the signal for the GDP-bound form might be difficult to detect by immunocytochemistry due to its relatively small population. We also had difficulties identifying defects in the distribution of Rabin8 in BAG6-suppressed cells (Appendix Fig S8B). Although we were disappointed not to see any defects in their distribution, we included these observations as Appendix Fig S8 and added relevant sentences as follows.

Appendix Figure Legends: p.5, line 94-100.

“Appendix Figure S8. In relation to Fig EV4A, no detectable abnormality of Rab8a and Rabin8 localizations in BAG6-suppressed cells. Immunostaining of the endogenous Rab8a (A, C, shown as green) in HeLa (A) and CHO (C) cells. Immunostaining of the endogenous Rabin8 in HeLa cell (B, shown as green). The 550 positions of the nucleus are shown by Hoechst staining (blue). All cells were transfected with a series of siRNA constructs as indicated. Scale bars: 10 μ m.”

2.

Fig EV1A is missing a positive control for polyubiquitin stain to show that the antibody is working and what would be expected should polyubiquitination occur. According to this comment, we added a positive control experiment showing that MG-132-induced aggresomes, which are well-characterized, polyubiquitin-positive, large perinuclear protein aggregates, were stained by an anti-polyubiquitin FK2 antibody under identical experimental condition. This result clearly shows that the FK2 antibody used in this study works in our conditions. Accordingly, we added these data to Appendix Fig S1A.

Appendix Figure Legends: p.1, line 22-p.2, line 26.

“(A) HeLa cells were treated with MG-132 (10 μ M) for 12 h, and fixed cells were stained with anti-polyubiquitin FK2 antibody (green) under identical experimental condition to Fig EV1A. Aggresome, a perinuclear insoluble aggregate induced by MG-132-treatment, was stained by anti-polyubiquitin FK2 antibody (indicated by an arrow). The positions of the nucleus are shown by Hoechst staining (blue). Scale bar: 10 μ m.”

3-1.

Rab8a KD is shown to cause similar defects in TfnR localisation to BAG6 KD. It would be interesting to see if Rab8 knockdown induced the same abnormal distribution of Golgi apparatus markers, Ptch1 and Rab7 too to clarify which of the presented phenotypes relate to Rab8 and which to the other Rabs.

In accordance with this important suggestion, we examined whether Rab8a knockdown induced the abnormal distribution of the endosomal protein Ptc1 in HeLa cells, and the *trans*-Golgi apparatus marker Stx6 and *cis*-Golgi markers GS28 and GM130 in CHO cells. As a result, we found that Rab8a depletion caused defects in the localization of all of these organelle-associated marker proteins. For example, the distribution of Ptc1 protein was altered greatly to perinuclear compartments and reduced cytoplasmic vesicular dots in Rab8a-depleted cells (Fig EV1B), an almost identical phenotype to that observed in BAG6-depleted cells (Figs 1B and EV1A). An abnormally dispersed distribution of Stx6 was observed in Rab8a-depleted CH 585 O cells, which was nearly indistinguishable to the case in BAG6-depleted cells (Fig EV4B). GS28 and GM130 signals were also dispersed throughout the perinuclear region of the cytoplasm, an overlapping phenotype to that observed in BAG6-depleted cells (Fig EV4C). As we had previously demonstrated in Fig 1A, the distribution of the recycling endosomal protein TfnR was also affected similarly by BAG6 or Rab8a depletion. Although we are unable to discriminate the phenotypes associated with Rab8a and the other Rabs conclusively, these observations support our original view that the function of BAG6 is closely linked with that of Rab8a protein. Accordingly, we have added the new Figures EV1B, EV4B, EV4C and relevant sentences in the Results and Figure Legend sections.

Thank you very much for providing us with an opportunity to strengthen our results.

p.6, line 118-120.

“The defective distribution of Ptc1 was also observed following Rab8a depletion (Fig EV1B), which was a similar phenotype to that observed in BAG6-depleted cells (Fig 1B and Fig EV1A).”

p.16, line 315-317.

“The abnormally dispersed distribution of Stx6 was also observed in Rab8a-depleted CHO cells, which was nearly indistinguishable to the case in BAG6-depleted cells (Fig EV4B).”

p.17, line 322-325.

“In BAG6 knockdown cells, however, their signals were separated with less

co-localization throughout the perinuclear region of the cytoplasm (Fig 7B, right panel; compare the red and green signals), a similar phenotype to that observed in Rab8a-depleted cells (Fig EV4C).”

p.52, line 1031-1032.

“(B) Immunostaining of Ptc1 (green) in HeLa cells that were treated with or without *Rab8a* siRNA. See also Appendix Fig S1D. Scale bar: 10 μ m.”

Appendix Figure legend; p.2, line 33-35.

“(D) Immunostaining of the Ptc1 (green) in HeLa cells that were treated with *Rab8a* siRNA (left), *BAG6* siRNA (center) and their combination (right). The positions of the nucleus are shown by Hoechst staining (blue). See also Fig 1B and Fig EV1B. Scale bars: 10 μ m.”

3-2.

Perhaps the authors could also check if double depletion of Bag6 and Rab8 could rescue some of the observed membrane sorting defects.

In accordance with this convincing suggestion, we tried to knockdown BAG6 and Rab8 simultaneously, and examined whether the double depletion of BAG6 and Rab8 could rescue some of the observed membrane sorting defects in BAG6 knockdown cells. As shown in Fig EV4C, it was difficult for us to see a difference between the phenotypes of BAG6 knockdown and BAG6/Rab8a double depletion. Similarly, we did not see an obvious rescue effect in the case of Ptc1 (Appendix Fig S1D). We reasoned that Rab8a knockdown not only down-regulates the harmful accumulation of the GDP-bound form (a BAG6 target) but also depletes the major active form of Rab8a (please note that the GTP-bound form is not a target of BAG6). Accordingly, we presented the results of this experiment in the new Fig EV4C and Appendix Fig S1D in our revised manuscript.

p.54, line 1078-p.55, line 1083.

“(C) Comparison of the defects observed in the distribution of the ER-Golgi SNARE protein GS28 (green) and the *cis*-Golgi marker GM130 (red) with siRNAs (*cBAG6* siRNA#2 or *RAB8a* siRNA#1 and their combination) in CHO cells. GS28 and GM130 signals were dispersed throughout the perinuclear region of the cytoplasm in Rab8a knockdown cells, a similar phenotype to that observed in BAG6-depleted cells. Scale bar: 10 μ m.”

Appendix Figure Legends: p.2, line 33-35.

“(D) Immunostaining of the Ptc1 (green) in HeLa cells that were treated with *Rab8a* siRNA (left), *BAG6* siRNA (center) and their combination (right). The positions of the nucleus are shown by Hoechst staining (blue). See also Fig 1B and Fig EV1B. Scale bars: 10 μ m.”

4.

Authors show in Fig.3 that the T22N mutant, but not active Rab8a, is ubiquitinated. Recently it has been described that BAG6 dependent ubiquitylation process is mediated by VCP-UBXN1 complex (Ganji et al.2018). It would be interesting to address if Rab8a ubiquitylation is also dependent on this complex or if Bag6 acts independently in this case. At the very least perhaps it is worth speculating on and including the reference.

According to this insightful suggestion, we examined the effects of the depletion of UBXN1 and VCP/p97/CDC48. As anticipated, BAG6 depletion reproducibly stimulated the accumulation of the Rab8a T22N mutant, while its polyubiquitination was greatly compromised (Data for Referee #3). In the case of VCP knockdown, cell viability was greatly reduced, probably due to its essential function in viability; thus, we found it difficult to conclude whether VCP/p97/CDC48 is indeed indispensable for the polyubiquitination of Rab8a. In the case of UBXN1 knockdown, we did not see reproducible effects on the polyubiquitination of Rab8a protein, which was sometimes decreased and other times not changed. We are sorry to say that we could not obtain conclusive results with this experiment. Accordingly, we added relevant explanations in the Discussion section as follows and cited the reference Ganji et al. (2018), as

suggested by Referee #3:

Data for Referee #3:

Polyubiquitination of Rab8a T22N in *UBXN1*, *VCP*, *BAG6* knockdown cells. Flag-Rab8a (T22N) immunoprecipitates were blotted with an anti-polyubiquitin antibody (FK2). In this representative result, ubiquitin co-precipitation seemed to be reduced in *ubxn1* knockdown cells, but the observation varied between experiments; thus, it was hard for us to conclude anything at this moment. Please note that *BAG6* knockdown stimulated the accumulation of Rab8a T22N, while its polyubiquitination was decreased reproducibly.

p.20, line 383-385.

“Recently, it was reported that some BAG6 client proteins are degraded by the UBXN1-mediated pathway [82]. Therefore, it might be interesting to examine this possibility with Rab8a T22N as a substrate.”

p.42, line 829-831.

“82. Ganji R, Mukkavalli S, Somanji F, Raman M (2018) The VCP-UBXN1 complex mediates triage of ubiquitylated cytosolic proteins bound to the BAG6 complex. *Mol Cell Biol* doi: 10.1128/MCB.00154-18”

Manuscript criticisms:

1.

Fig1 and EV1 please make the images brighter. The DAPI staining is missing or very dim in Fig1A.

In accordance with this suggestion, we have improved brightness of Fig 1B and Fig EV1A (We are sorry to say that we could not enhance DAPI image in Fig 1A). We hope that the changes have improved these figures greatly and that Figures are now acceptable for publication.

2.

In all the figures with IPs BAG6 is annotated as S-BAG6. Please specify the meaning of s-BAG6 in the method section or figure legend.

S-BAG6 stands for N-terminally S-tagged BAG6 protein. The S-tag is a 15 amino acid-long peptide (KETAAAKFERQHMDs), and fusion proteins with S-tag can be efficiently precipitated by non-IgG protein (S-protein)-conjugated agarose beads (69704, Merck KGaA, Germany). Accordingly, we have added sentences describing the S-tag used in the Materials and Methods section as follows:

p.26, line 505-508.

“For some experiments, we precipitated S-tagged BAG6 from cell lysates. The S-tag is a 15 amino acid-long peptide (KETAAAKFERQHMS), and fusion proteins with an N-terminal 2x S-tag can be precipitated efficiently by non-IgG affinity protein (S-protein)-conjugated agarose beads (69704; Merck KGaA, Germany).”

p.45, line 889-890.

“S-BAG6 stands for N-terminally S-tagged BAG6 protein (see the Materials and Methods).”

3.

The schematic mentioned in the figure legend for BAG6 mutants is missing (Fig3D). Please add this to the figure.

Thank you very much for bringing this to our attention. We apologize for this oversight and are grateful to Referee #3.

p.46, line 914-917.

“A schematic representation of the BAG6-truncated proteins used in this experiment is shown in the upper panel. Numbers denote the corresponding amino acids of mammalian BAG6. Positions of UBL, BUILD, and DUF3538 domains, which are all linked to hydrophobicity recognition by BAG6 [37], are indicated.”

4.

One of the most interesting pieces of data is that Bag6 specifically binds to the hydrophobic region present in the Switch I of the GTPase domain (Fig.4). This region is formed in Rab8a by three hydrophobic amino acids. Interestingly these hydrophobic amino acids are absent in Rab7 which does not interact with BAG6 - it would be informative to see the alignment of all the Rab proteins in Fig.6.

According to this kind comment, we have provided amino acid sequence alignment of the Switch I region of all Rab proteins listed in Fig 6 as Appendix Fig S4. In addition to this information, we examined the hydrophobicity profile of Rab family proteins with a Kyte-Doolittle hydrophobicity plot (provided as Appendix Fig S5). As you will see, BAG6-phile Rab species (such as Rab8a, 3a, 2b, and 6b) tended to possess a higher hydrophobicity peak in the Switch I region (boxed as a red rectangle in Appendix Fig S5) compared with Rab proteins with weak BAG6 interactions (such as Rab7a, 12, 11b, and 4a). Accordingly, we provided two new Figures as Appendix Figs S4 and S5 and relevant sentences in the Appendix Figure Legends as follows.

p.16, line 301-304.

“Amino acid sequence alignments of the Switch I regions (Appendix Fig S4) and Kyte-Doolittle hydrophobicity plots (Appendix Fig S5) suggest that BAG6-phile Rab species tended to possess a higher hydrophobicity peak in the Switch I region compared with Rab proteins with weak BAG6 interactions.”

Appendix Figure Legend, p.3, line 63-p.3, line 69.

“Appendix Figure S4. In relation to Figs 4A and 6B, amino acid sequence alignments of the Switch I region of the Rab proteins. The distribution of the hydrophobic residues in the Switch I region is indicated in color. The hydrophobic residues that are frequently found in BAG6-phile Rab proteins are shown in red (indicated by red arrowheads), while the other hydrophobic residues are shown in orange.”

Appendix Figure Legend, p.4, line 71-76.

“Appendix Figure S5. In relation to Appendix Fig S4, Kyte-Doolittle hydrophobicity plots of BAG6-phile Rab proteins and that with weak BAG6 interaction. The hydrophobicity peak within Switch I region of respective Rab proteins are indicated as red rectangle. The horizontal axis numbers denote the corresponding amino acid positions in these proteins.”

5-1.

Linked to the above point, discussion about why some Rabs bind BAG6 and some do not would be useful and interesting. For example, is it the structure of

the switch I region or is it Rabs involved in particular pathways/organelles or is it not predicabile.

Thank you very much for your stimulating idea about the mechanism by which some Rabs bind BAG6 and some do not. As discussed in the previous sections, BAG6-phile Rab species tended to possess higher hydrophobicity in the Switch I region, suggesting that this could be one of the factors that determined the affinity of Rab proteins with BAG6. This view is further strengthened by our new experiments using chimeric Rab8a protein with the Rab7-type Switch I sequence (Fig EV3A, B). This experiment revealed that such a chimeric Rab8a had reduced affinity with BAG6 compared with wild type Rab8a, suggesting the importance of the Switch I sequence. However, it seems likely that the Switch I sequence is not the sole determinant of BAG6 affinity, since a Rab7 GDP mutant (Rab7a T22N) also showed increased affinity for BAG6 (Appendix Fig S7), although its affinity was less than that of Rab8a T22N. Collectively, we consider that there are at least two critical factors that determine BAG6 affinity; one is the Switch I sequence (exposed hydrophobicity) and the other is the ratio of the GDP-/GTP-bound forms of respective Rab species, which are greatly influenced by their specific GEF/GAP interactions in the cells. Indeed, our experiments showed that reducing Rab8a GEF (Rabin8) greatly enhanced the affinity of Rab8a for BAG6 (Fig 2F). Accordingly, we discussed these possibilities in the Discussion section as follows.

Thank you very much for providing us with an opportunity to discuss this useful and interesting possibility.

p.13, line 254-256.

“In addition, the “8a-7a chimera” T22N protein with Rab7-type Switch I sequence showed reduced affinity with BAG6 compared to the case for Rab8a T22N (Fig EV3A, B).”

p.20, line 386-p.21, line 401.

“The mechanism by which some Rab proteins bind BAG6 and others do not is an interesting issue. As shown in Fig 4A and Appendix Figs S4 and S5, BAG6-phile Rab species tended to possess higher hydrophobicity in the Switch I region, suggesting that this could be one of the factors that determine their affinity with BAG6. This view was further strengthened by our experiments using chimeric Rab8a T22N protein with the Rab7-type Switch I sequence (8a-7a chimera, Fig EV3). This experiment revealed that such a chimeric Rab8a (T22N) protein had reduced affinity with BAG6 compared to Rab8a (T22N), suggesting the importance of the Switch I sequence. However, it seems likely that the Switch I sequence is not the sole determinant for BAG6 affinity, since a Rab7 GDP mutant (Rab7a T22N) also showed increased affinity for BAG6, although its affinity was less than that of Rab8a T22N (Appendix Fig S7). Collectively, we consider there are at least two critical factors that determine the affinity of Rab protein for BAG6: one is the Switch I sequence (exposed hydrophobicity) and the other is the ratio of the GDP-/GTP-bound forms of respective Rab species, which is greatly influenced by their specific GEF/GAP interactions in the cells. Indeed, our experiments showed that reducing Rab8a GEF (Rabin8/Rab3IP) activity greatly enhanced its affinity for BAG6 (Fig 2F).”

p.53, line 1057-p.54, line 1064.

“**Figure EV3.** In relation to Fig 4, the Switch I region of Rab8a is critical for its interaction with BAG6. (A) Schematic representation of the Switch I region of the Rab8a-Rab7a chimeric protein. The amino acid residues 33-45 (boxed region) of full-length Rab8a were substituted with those of Rab7a and this mutant protein was designated as the “8a-7a chimera”. The numbers denote the corresponding amino acids of human Rab8a and Rab7a. (B) The 8a-7a chimera protein with the T22N mutation showed greatly reduced affinity with BAG6 compared with the case in Rab8a T22N.”

Appendix Figure Legend: p.4, line 88-p.5, line 92.

“Appendix Figure S7. In relation to Fig EV3, Rab7a GDP mutant show increased affinity for BAG6 compared to the case in its wild type. Flag-tagged WT Rab8a, WT Rab7a and their TN mutant derivatives (GDP-bound forms) were immunoprecipitated and probed with an anti-BAG6 antibody. MG-132 (10 μM) was included in the cell

culture for 4 h.”

5-2.

Is it Rabs involved in particular pathways/organelles or is it not predicable.

We did not see any correlation between the retrograde/anterograde trafficking function of Rab proteins and the order of BAG6 affinity. Accordingly, we have added a following sentence.

p.16, line 304-306

“We did not observe any correlation between the retrograde/anterograde trafficking function of Rab proteins and the order of BAG6 affinity.”

6.

Fig. 5B. Quantification of band density at all time points would be more useful to see the extent of degradation over time.

In accordance with this request, we quantified all of the blot signals in the CHX chase experiments. In this revision, we have repeated this experiment, and confirmed that Rab8a T22N protein was reproducibly stabilized under BAG6 siRNA knockdown. Accordingly, we substituted the previous Fig 5B with the newly obtained quantified data at all time points to see the extent of degradation over time.

p.48, line 955-959.

“(B) Anti-Flag blot signals in the control or BAG6 siRNA-treated cells were quantified, and relative signal intensities after CHX addition were calculated. The value of the Flag signal at 0 h was defined as 1.0. Note that all signal intensities of the Flag-tag were 855 normalized by that of actin, a loading control, in each sample. The graph represents the mean \pm S.E. calculated from 6 independent biological replicates.”

7-1.

Fig.6 and the discussion of Fig.6 is not systematic in how it deals with the Rab proteins and there are inconsistencies. For example many 860 of the Rab proteins shown on the blot are then not quantified on the graph ie 35, 6a, 5b.

We would like to apologize for the inconsistencies in the previous Fig 6. The reason why we could not show quantified data for Rab35, 6a, 5b, and 3a was due to the small number of experiments performed (less than 3 independent biological replicates) with these Rab species. Accordingly, we have repeated this experiment with each Rab protein, and now we have provided quantified data for Rab35, 6a, 5b, and 3a. Therefore, we replaced the original Fig 6B with the revised version and added relevant sentences as follows.

p.50, line 983-987.

“The number of independent biological replicates was as follows: Rab1a, n = 4; Rab1b, n = 5; Rab2b, n = 3; Rab3a, n = 4; Rab4, n = 4; Rab5a, n = 3; Rab5b, n = 4; Rab6a, n = 5; Rab6b, n = 4; Rab7a, n = 3; Rab8a, n = 14; Rab9b, n = 4; Rab10, n = 6; Rab11a, n = 3; Rab11b, n = 3; Rab12, n = 2; Rab13, n = 6; Rab14, n = 3; Rab15, n = 5; Rab35, n = 875 3.”

7-2.

Also for some Rabs, the blots presented are not representative of the quantification, for example, a large amount of Rab10 and 1b seems to bind to Bag6 in the blot but not the graph. This may be an artefact of the normalisation and this needs to be looked at more carefully.

Thank you very much for your suggestion. The reason why the previous quantification of some Rabs, such as Rab10 and 1b, did not correspond exactly to the representative blot was largely due to the variation of the results in each experiment. Accordingly, we have repeated the experiments, re-quantified the data, and substituted the original graph with the new data in this revised manuscript.

Thank you very much for providing us with an opportunity to strengthen our results. We greatly appreciate valuable suggestions from Referee #3 to improve our study and

that our manuscript is now acceptable for publication in *EMBO Reports*.
Thank you very much for your kind support.

2nd Editorial Decision

18 January 2019

Thank you for the submission of your revised manuscript to EMBO reports. It was evaluated again by referees 2 and 3 and we have now received their reports (copied below).

As you will see, both referees are very positive about the study and support publication without further revision.

Browsing through the manuscript myself, I noticed a few things that we need from the editorial side before we can proceed with the official acceptance of your study.

REFEREE REPORTS

Referee #2:

My minor comments have been addressed.

Referee #3:

The authors have done a lot of work to address our comments and I am satisfied that it has improved and is now suitable for publication.

2nd Revision - authors' response

25 January 2019

The authors performed all minor editorial changes.

Corresponding Author Name: Prof. Hiroyuki Kawahara

Manuscript Number: EMBOR-2018-46794V2